# Multi-omic phenotyping reveals host-microbe responses to bariatric surgery, glycaemic control and obesity

Nicholas C. Penney [1,2✉], Derek K. T. Yeung [1,2], Isabel Garcia-Perez [1], Joram M. Posma [1,3], Aleksandra Kopytek[1], Bethany Garratt[2], Hutan Ashrafian [1,2], Gary Frost [1], Julian R. Marchesi[1], Sanjay Purkayastha[2], Lesley Hoyles[1,4], Ara Darzi [2,5] & Elaine Holmes[1,6✉]

## Abstract

**Background** Resolution of type 2 diabetes (T2D) is common following bariatric surgery, particularly Roux-en-Y gastric bypass. However, the underlying mechanisms have not been fully elucidated.

**Methods** To address this we compare the integrated serum, urine and faecal metabolic profiles of participants with obesity ± T2D ($n = 80$, T2D $= 42$) with participants who underwent Roux-en-Y gastric bypass or sleeve gastrectomy (pre and 3-months post-surgery; $n = 27$), taking diet into account. We co-model these data with shotgun metagenomic profiles of the gut microbiota to provide a comprehensive atlas of host-gut microbe responses to bariatric surgery, weight-loss and glycaemic control at the systems level.

**Results** Here we show that bariatric surgery reverses several disrupted pathways characteristic of T2D. The differential metabolite set representative of bariatric surgery overlaps with both diabetes (19.3% commonality) and body mass index (18.6% commonality). However, the percentage overlap between diabetes and body mass index is minimal (4.0% commonality), consistent with weight-independent mechanisms of T2D resolution. The gut microbiota is more strongly correlated to body mass index than T2D, although we identify some pathways such as amino acid metabolism that correlate with changes to the gut microbiota and which influence glycaemic control.

**Conclusion** We identify multi-omic signatures associated with responses to surgery, body mass index, and glycaemic control. Improved understanding of gut microbiota - host co-metabolism may lead to novel therapies for weight-loss or diabetes. However, further experiments are required to provide mechanistic insight into the role of the gut microbiota in host metabolism and establish proof of causality.

### Plain Language Summary

Weight-loss surgery is a highly effective treatment of type 2 diabetes in people with obesity. Interestingly, the improvement in diabetes after weight-loss surgery occurs before any significant weight-loss. Through better understanding of this metabolic improvement, weight-loss surgery provides a unique avenue to identify novel ways of treating diabetes and obesity. Here we combine measurements of metabolism, gut bacteria and diet in people with obesity, with or without type 2 diabetes and in patients before and after weight-loss surgery. We have used these data to identify changes associated with weight-loss surgery, obesity and diabetes. Improved understanding of the mechanisms behind these changes, including how changes to gut bacteria influence metabolism, may lead to new treatments for weight-loss or diabetes.

[1] Department of Metabolism, Digestion and Reproduction, Faculty of Medicine, Imperial College London, London SW7 2AZ, UK. [2] Department of Surgery and Cancer, Faculty of Medicine, Imperial College London, London W2 1NY, UK. [3] Health Data Research UK, London NW1 2BE, UK. [4] Department of Biosciences, Nottingham Trent University, Nottingham NG11 8NS, UK. [5] Institute of Global Health Innovation, Imperial College London, London W2 1NY, UK. [6] Centre for Computational & Systems Medicine, Health Futures Institute, Murdoch University, Perth, WA 6150, Australia. ✉email: n.penney@imperial.ac.uk; elaine.holmes@imperial.ac.uk

The global epidemic in obesity and associated disease states carries a significant health and economic burden. The gut microbiota (GM) has been implicated as a contributing factor in a number of these diseases, including obesity and type 2 diabetes (T2D)[1–3]. Faecal microbiota transplant experiments in obesity[3] and T2D[3,4] have shown that this relationship is causal, but these studies have failed to fully unravel the complex mechanisms behind this observation, further complicated by the fact that each individual's GM is unique and subject to redundancy in its metabolic function[5]. Therefore, there is a need to move beyond simply profiling the composition of GM communities in order to understand the true nature of host-microbe relationships.

Surgical procedures such as Roux-en-Y gastric bypass (RYGB) and vertical sleeve gastrectomy (VSG) achieve sustainable weight-loss in obesity[6]. Importantly, they are also highly successful in the resolution of obesity-related co-morbidities including T2D[7]. These metabolic outcomes are achieved through both weight-dependent and, interestingly, weight-independent mechanisms[8]. Weight-independent effects occur because bariatric surgery, particularly RYGB, induces a complex system-wide metabolic effect, including modification of the GM-host metabolic axis[9]. The overwhelming disruption to the GM caused by bariatric surgery is only just being defined, as is its functional importance. To date, few studies have explored longitudinal host-microbe interactions in human cohorts following bariatric surgery, with most studies focussing on either the microbiota or the metabolome[10,11]. Multiple mechanisms for the GMs contribution to achieving weight-loss and metabolic improvement post-surgery have been hypothesised, including: reduced energy harvest of non-digestible food types such as complex carbohydrates; reduced gut permeability leading to decreased systemic inflammation; and alterations in microbe-host co-metabolites such as bile acids (BAs), amino acids (AAs) and short-chain fatty acids (SCFAs)[10,12].

Bariatric surgery provides a unique opportunity to unravel these complex host-microbe interactions through phenotyping before and after intervention to reduce the impact of inter-individual variability. Here we have performed multi-platform profiling, to establish changes in the host-microbe interactions in volunteers with obesity ± T2D and in individuals undergoing bariatric surgery with and without T2D to identify dysregulated pathways in T2D that are functionally restored after bariatric surgery. First, we have compared differences in GM-host co-metabolism in participants with T2D compared to individuals without diabetes at baseline to ascertain which metabolites were associated with glycaemic control. Next, we profiled subgroups of patients undergoing RYGB or VSG to evaluate changes in GM-host co-metabolism following these contrasting interventions and assessed their impact on glycaemic control, taking into account intervention-dependent changes in eating behaviour.

Our results provide a comprehensive atlas of host-gut microbe responses to bariatric surgery, weight-loss and glycaemic control at the systems level. We find minimal overlap between metabolites associated with body mass index (BMI) and those associated with glycaemic control, consistent with weight-independent mechanisms of T2D resolution. Further, we establish multiple associations between the gut microbiota and host metabolites. However, further experiments are required to provide mechanistic insight into the role of the gut microbiota in host metabolism and establish proof of causality.

## Methods
### Experimental model and participant details
*Recruitment.* Patients referred for consideration of bariatric surgery who were obese (BMI > 30 kg/m²), aged ≥ 18, had failed efforts at lifestyle modification and dieting and were willing to comply with the trial protocol were recruited prospectively to this observational study, with volunteers providing written consent. Volunteers with type 2 diabetes (HbA1c ≥ 48 mmol/mol or treated), impaired glucose tolerance (HbA1c 42–47 mmol/mol) and without diabetes were eligible for recruitment.

Patients who had previously undergone bariatric or major abdominal surgery, were or intended to become pregnant during the trial period, or took long-term antibiotics were excluded. Major abdominal surgery included patients who had undergone small or large bowel resection, liver, pancreatic, splenic or stomach surgery, as these could influence the gut microbiota and/or the patient's metabolic state. Patients that had previously had an appendicectomy, cholecystectomy or hernia repair were not excluded.

The study protocol and sample collection instructions were co-developed with patient representatives to help reduce the study burden for patients. To improve patient compliance sample collection occurred at the time of patients' usual hospital appointments preoperatively and at 3-months post-procedure. A small exploratory cohort were also sampled at 1-year post-procedure and are reported in Supplementary Figs. 4–9, Supplementary Tables 6–8 and Supplementary Data 6–8.

*Metabolic surgery.* Participants underwent Roux-en-Y Gastric Bypass (RYGB) or Vertical Sleeve Gastrectomy (VSG) surgery at a National Health Service (NHS) University Teaching Hospital in London, UK. A single dose of 1.2 g intravenous co-amoxiclav was given during induction of anaesthesia (clindamycin if penicillin allergic). See Supplementary Note 1 for further details of the perioperative dietary advice given and surgical techniques used.

*Regulatory approvals.* The study received NHS Research Ethics Committee (15/ES/0026) approval and was registered with ClinicalTrials.gov (NCT02421055).

*Clinical, demographic and dietary data collection.* Participants were assessed at the above time points for: (1) anthropometric & physiological measurements, (2) demographic details, (3) biochemical parameters including glycated haemoglobin (HbA1c), (4) oral-hypoglycaemic, insulin and other medication use, length of diabetes diagnosis and other co-morbidities.

Diet assessment: An online self-reported 24-hour dietary recall questionnaire (www.myfood24.org) was utilised to capture detailed dietary intake information from patients at each of the study time points so that changes in diet following surgery could be accounted for in the analysis. Three 24-hour recall questionnaires were completed at each time point. Participants were able to pick from a selection of pictures of corresponding foods to accurately ascertain portion quantities. Collected dietary information was used to calculate Alternative Healthy Eating Index 2010 (AHEI-2010) scores as described previously[13]. In brief, scores of 0-10 were given for 11 components (maximum score 110). High scores were given for a high intake of vegetables, fruit, nuts and legumes, whole grains, long chain omega-3 fats and polyunsaturated fats, moderate intake of alcohol and low intake of sugar, sweetened drinks and fruit juice, red and processed meat, trans-fat and sodium.

Sample collection: Serum ($n = 156$), faecal ($n = 80$) and 24 h urine ($n = 83$) samples were collected preoperatively and at 3-months postoperatively (serum = 49, faecal = 27, urine = 30) in a non-fasted state (see Supplementary Fig. 1). Urine samples were collected in sterile containers over a 24-hour period finishing at 9am on the day of the study visit. Stool samples were

collected using a Faecotainer® collection kit and stored on an ice pack provided, as close as possible to but not more than 6 h before the study visit. Serum was collected using red top BD Vacutainer® serum tubes (no additive) and processed according to manufacturer guidelines. After collection, samples were aliquoted and stored at −80 °C until analysis. Prior to freezing, a separate aliquot of homogenised stool was used to generate faecal water as follows: approximately 10 g of stool was added to 4 parts HPLC grade $H_2O$ (g/ml), vortexed at 2850 rpm for 15 min then centrifuged at 10,000 × g for 15 min at 4 °C. The resulting supernatant (faecal water) was frozen at −80 °C until analysis[14].

### Analysis methods

#### $^1$H-NMR spectroscopic analysis of metabolites

Sample preparation: Urine and faecal water samples were prepared for analysis by $^1$H-NMR spectroscopy as follows: frozen samples (−80 °C) were thawed, vortexed and then centrifuged at 1600 × g for 10 min to remove particulates and precipitated proteins. Faecal water supernatant was further filtered through Micro centrifuge filters (0.45 μm Nylon, Costar) at 16,000 × g for 15 min at 4 °C. 540 μL of each sample was mixed with 60 μL of 1.5 M $KH_2PO_4$ buffer (pH 7.4, 80% $D_2O$) containing 1 mM of the internal reference standard, 3-(trimethylsilyl)-[2,2,3,3,-$^2H_4$]-propionic acid (TSP) and 2 mM sodium azide ($NaN_3$), as described previously[15].

After thawing, serum samples were centrifuged at 12,000 × g for 5 min at 4 °C. Subsequently, 300 μL of serum was mixed with 300 μL of 0.075 M $NaH_2PO_4$ buffer (pH 7.4) containing 0.8 mM of the internal reference standard, 3-(trimethylsilyl)-[2,2,3,3,-$^2H_4$]-propionic acid (TSP) and 3.1 mM sodium azide ($NaN_3$), as described previously[15].

$^1$H-NMR spectroscopy: $^1$H-NMR spectroscopy was performed at 300 K on Bruker 600 MHz (urine and serum) and 800 MHz (faecal water) spectrometers (Bruker Biospin) using the following standard one-dimensional pulse sequence: RD – $g_{z1}$ – 90° – $t_1$ – 90° – $t_m$ – $g_{z2}$ – 90° – ACQ[15]. The relaxation delay (RD) was set at 4 s, 90° represents the applied 90° radio frequency pulse, interpulse delay ($t_1$) was set to an interval of 4 μs, mixing time (tm) was 10 ms, magnetic field gradients ($g_{z1}$ and $g_{z2}$) were applied for 1 ms and the acquisition period (AQA) was 2.7 s. Water suppression was achieved through irradiation of the water signal during RD and $t_m$. For the urine samples, each spectrum was acquired using 4 dummy scans followed by 32 scans while faecal spectra were acquired using 256 scans and 4 dummy scans and collected into 64 K data points. A spectral width of 12,000 Hz was used for all the samples. Prior to Fourier transformation, the free induction decays (FIDs) were multiplied by an exponential function corresponding to a line broadening of 0.3 Hz. Serum samples were analysed by $^1$H-NMR spectroscopy using the standard one-dimensional pulse sequence described above and Carr-Purcell-Meiboom-Gill (CPMG) one dimensional pulse sequences. CPMG was used to attenuate broad, interfering peaks from lipids and proteins present in serum. The CPMG pulse sequence had the form RD – 90° – (t – 180° – t) n – ACQ. The acquisition parameters were set using the same settings as the standard 1D pulse sequence, with the spin-echo delay (t) set at 0.3 ms and 128 loops (n) performed. Continuous wave irradiation was applied at the water resonance frequency during the relaxation delay (RD).

Pre-processing: $^1$H-NMR spectra were automatically corrected for phase and baseline distortions and referenced to the TSP singlet at δ 0.0 using TopSpin 3.1 software. Spectra were then digitized into 20 K data points at a resolution of 0.0005 ppm using an in-house MATLAB R2014a (Mathworks) script. Subsequently, spectral regions corresponding to the internal standard (δ −0.5 to 0.5) and water (δ 4.6–5) peaks were removed. In addition, the region containing urea (δ 5.4–6.3) was removed from the urinary and serum spectra due to its tendency to cross-saturate with the suppressed water resonance. All spectra were normalised using median fold change normalisation using the median spectrum as the reference[16].

*Quantitative bile acid analysis.* Quantitative analysis of 57 bile acids was performed using an established technique[17]. The method was adapted for analysis of bile acids in faecal samples.

Sample preparation: Bile acids were extracted from serum using the following method: 100 μL of serum was vortexed with 280 μL of MeOH. Samples were centrifuged at 14,000 × g for 15 min at 4 °C, followed by incubation at −20 °C for 20 min. Internal standards (16 deuterated bile acids) were added to the supernatant at a final concentration of 50 nM.

Bile acids were extracted from faecal samples using the following method. Faecal samples were first freeze-dried. 100 mg of freeze-dried material was then placed in microtubes with 1 ml of 2:1:1 $H_2O$: Acetonitrile (ACN): Isopropanol (IPA) and approximately 50 mg of 1 mm Zirconia beads. This underwent 3 × 30 seconds bead beating and a Biospec bead beater followed by centrifugation at 16,000 × g for 20 min at 4 °C. The supernatant was further filtered through Micro-centrifuge filters (0.45 μm Nylon, Costar) at 16,000 × g for 15 min at 4 °C. To ensure bile acid concentrations were within the dynamic range of the machine extracts were diluted 1:25 and 1:200 prior to analysis using the $H_2O$:ACN:IPA mix. A mixture of internal standards (16 deuterated bile acids) was added to the filtered supernatant at a final concentration of 50 nM.

LC-MS machine conditions: BA analysis was performed using an ACQUITY ultra-performance liquid chromatography (UPLC) coupled to a Xevo triple quadrupole (TQ-S) mass spectrometer.

For liquid chromatography, an ACQUITY BEH C8 column (1.7 μm, 100 mm × 2.1 mm) was used at an operating temperature of 60 °C. The mobile phase solvent A consisted of a 1:10 ACN:$H_2O$, with 1 mM ammonium acetate and pH 4.15 adjusted with acetic acid. Mobile phase solvent B consisted of 1:1 ACN:IPA. The chromatographic gradient was as previously published[17].

Mass spectrometry was performed in negative ionisation mode (ESI-) using the following parameters: capillary voltage 1.5 kV, cone voltage 60 V, source temperature 150 °C, desolvation temperature 600 °C, desolvation gas flow 1000 L/hr, and cone gas flow 150 L/hr. 57 bile acid species (36 non-conjugated, 12 taurine conjugated, 9 glycine conjugated) were assayed using multiple reaction monitoring (MRM). The transitions for each bile acid and deuterated internal standard were set as previously published[17].

*Quantitative analysis of SCFAs and other carboxylic acids.* A total of five short/medium chain fatty acids, three methyl-branched SCFAs and two hydroxyl carboxylic acids were analysed by GC-MS using a method adapted from Moreau et al.[18].

Sample preparation: After defrosting and mixing, 100 μL of urine / serum was aliquoted with 500 μL of methyl *tert*-butyl ether (MTBE) with 100ppm of internal standard (methyl stearate) and 2 μL of HCL. This was vortexed and then shaken for 20 min. Following this samples were centrifuged at 10,000 × g for 5 min at 4 °C. Next, 90 μL of the polar phase was placed into a silanised vial and vortexed with 150 μL of derivatiser *N-tert-*

**Table 1 GC-MS conditions used to analyse SCFA and carboxylic acid compounds.**

| Compound | Quantifier (m/z) | Qualifier (m/z) | Collision energy (eV) |
|---|---|---|---|
| Acetate | 117 → 75 | 117 → 47 | 10 |
| Propionate | 131 → 75 | 131 → 47 | 10 |
| Butyrate / Isobutyrate | 145 → 75 | 145 → 43 | 10 |
| Valerate / Isovalerate | 159 → 75 | 159 → 57 | 12 |
| 2 Methylbutyrate | 159 → 75 | 159 → 57 | 12 |
| 2 Hydroxybutyrate | 147 → 73 | 147 → 45 | 20 |
| Caproate | 173 → 75 | 173 → 81 | 15 |
| Lactate | 147 → 73 | 147 → 45 | 20 |
| Methyl stearate (IS) | 87 → 55 | 87 → 59 | 10 |

butyldimethylsilyl-$N$-methyltrifluoroacetamide with 1% *tert*-butyldimethylchlorosilane (MTBSTFA + 1% TBDMSCI). This was then incubated for 45 min at 60 °C before aliquoting into silanised inserts for analysis.

The method was modified to account for higher levels of SCFA in stool. After defrosting, 100 mg of stool was aliquoted with 1,000 μL of MTBE with 100 ppm of internal standard (methyl stearate) and 4 μL of HCL. 30 μL of the polar phase was mixed with 150 μL of derivatiser.

GC-MS machine conditions: Derivatised samples were analysed by GC-MS with a Bruker triple quadrupole (TQ) GC-MS/MS. Helium was used as a carrier gas at a constant flow rate of 1.5 ml/min through the column. The injector temperature was 250 °C with a split ratio 1:10. The temperature of the oven was started at 40 °C and increased at the rate of 46 °C/min to 127 °C, 2 °C/min to reach 131 °C, 30 °C/min to reach 160 °C, then 50 °C/min to reach a final temperature of 300 °C. The transfer line to the mass spectrometer was set at 280 °C. Targeted analysis of the ten compounds and internal standard was performed in multiple reaction monitoring mode (MRN) using the settings outlined in Table 1.

*Quantitative serum metabolite analysis.* Quantitative analysis of other metabolites in serum samples, including amino acids, biogenic amines, acylcarnitines, phosphatidylcholines, lysophosphatidylcholines and sphingolipids was performed using the Biocrates AbsoluteIDQ® p180 kit, according to the manufacturer guidelines[19]. Samples were analysed using flow injection analysis (FIA)-MS/MS and LC-MS/MS for different metabolite groups.

Sample preparation: In total, 10 μL of serum sample / PBS / calibration / QC and 10 μL of the ISTD mix (except in blanks) was added to each well. This was dried for 30 min under nitrogen flow. Following this, 50 μL of the derivatization solution was pipetted into each well. The plate was covered and incubated for 20 min, then dried for 60 min under nitrogen flow. Next, 300 μL of extraction solvent was added to each well, shaken for 30 min at 450 rpm, then centrifuged for 2 min at 500 × g. For the LC-MS/MS 150 μL was added to 150 μl H$_2$O. For the FIA, 15 μL was added to 750 μL of FIA mobile phase. Both plates were shaken for 2 min at 600 rpm.

Machine conditions: Samples were analysed using a Waters I-Class UHPLC system and Waters Xevo TQ-S tandem mass spectrometer.

For FIA-MS/MS (direct infusion): the FIA mobile phase consisted of Biocrates Solvent I + 290 mL MeOH. A 2 min isocratic method was used, starting at 0.15 mL/min for 0.1 min, gradually decreasing to 0.03 mL/min at 1 min, increasing to 0.2 mL/min at 1.5 min, to 0.8 mL/min at 1.60 min, and finally decreasing to 0.15 mL/min at 1.95 min. MS settings were: capillary voltage 3.2 kV, cone voltage 10 V, source offset 50 V, source temp 150 °C, desolvation temp 620 °C, cone gas 150 L/H, desolvation gas 1000 L/H, collision gas flow 0.15 mL/min, probe position 5 mm.

For LC-MS/MS: A Waters Acquity UPLC BEHC18 1.7 μm 2.1 x 75 mm column was used. Mobile phase A: 1000 mL H$_2$O + 2 mL formic acid (FA), Mobile phase B: 500 mL ACN + 1 mL FA. Gradient elution was used; starting at a flow rate of 0.8 mL/min with 100% A for 0.45 min, then changing in a linear gradient to 85% A at 3.3 min, to 30% A at 5.9 min, to 100% B at 6.05 min, flow then increased in a concave gradient to 0.9 mL/min 100% B by 6.20 min, remaining at 0.9 mL/min 100% B until 6.42 min, before decreasing back in a concave gradient to 0.8 mL/min 100% B at 6.52 min. The mobile phase was then changed in a concave gradient from 100% B to 100% A between 6.52 and 6.7 min and remained at 100% A 0.8 ml/min until 7.3 min. MS settings were: capillary voltage 3.9 kV, cone voltage 20 V, source offset 50 V, source temp 150 °C, desolvation temp 350 °C, cone gas 150 L/Hr, desolvation Gas 650 L/Hr, collision gas flow 0.15 mL/min, probe position 7 mm.

Data were processed using targetlynx (Waters) and METIDQ (Biocrates; version Carbon) then exported as a CSV file for statistical analysis.

*Faecal metagenomic analysis*
DNA extraction: Faecal samples were stored at −80 °C prior to analysis. DNA was extracted using the MoBio PowerFaecal® DNA Isolation Kit, according to manufacturer's instructions. In brief, DNA was extracted from two separate 0.25 g aliquots of mixed whole stool samples at each analysis point. Samples were homogenised in 2 ml bead beating tubes containing garnet beads. Cell lysis of host and microbial cells was facilitated through both mechanical collisions between beads and chemical disruption of cell membranes. The reagent to precipitate non-DNA organic and inorganic material was then applied. Lastly, DNA was captured on a silica spin column, washed and eluted for downstream analysis. Quality control of DNA quality and quantity was assessed using an Agilent 4200 TapeStation.

Shotgun sequencing: Shotgun sequencing was performed using an Illumina HiSeq 4000 with paired-end 150 bp reads. Library preparation was undertaken using the NEBNext Ultra II DNA Library Prep Kit. 15 dual index barcodes (unique at both ends) were custom-designed and ordered from Integrate DNA Technologies (IDT®). Quality control of prepped libraries was performed using the Promega GloMax® and QuantiFluor® dsDNA systems. Each of the "uniquely dual-indexed" libraries were pooled and run on a single lane of the HiSeq4000. A mean of 6.89 Gb sequence data was acquired for each of 120 samples (median 6.84, range 3.88 – 14 Gb).

Processing of sequence data: There are known lane-swapping issues with the HiSeq 4000, leading to duplication of some sequencing reads. For this reason, fastq files for each sample were subject to de-duplication using FastUniq[20]. Sequencing data were then processed using the Scalable Metagenomics Pipeline (ScaMP)[21], (https://github.com/jamesabbott/SCaMP). In brief, raw sequence data were assessed for the presence of adapter sequences and trimmed using Trim Galore! (Babraham Bioinformatics) to remove low-quality bases (Q < 20) from the 3′ end of reads and discard trimmed reads shorter than 100 nt. Quality control of trimmed reads was performed using FastQC

(Babraham Bioinformatics). Reads that mapped with BWA-MEM[22] to human genome (hg19) were removed from read pairs, as ethical permission is not available for use of human data derived from metagenomes. Remaining reads were assumed to be microbial (bacteria, archaea, virus, fungi, protozoa) and processed further. Trimmed sequence data with human reads removed have been deposited with GenBank, EMBL and DDBJ databases under the BioProject accession number PRJNA473348.

MetaPhlAn 2.6[23,24] was used to determine the bacterial and archaeal taxonomic composition/abundance for each sample. Metagenome assembly was carried out in two rounds using metaSPAdes 3.11.0[25], with an initial independent assembly carried out for each sample. Unassembled reads were then pooled and subjected to a second round of assembly to improve the representation of low-abundance sequences. Taxa were normalised to relative abundance for downstream analyses.

Ab-initio gene prediction was carried out using MetaGeneMark[26,27]. The resulting predictions were translated, and the protein sequences clustered using the cluster-fast method of UCLUST[28], with a 95% identity cut-off. Centroid sequences from each cluster were used to form a non-redundant gene catalogue used for downstream analyses. Gene abundance in each sample was determined by alignment of the reads using BWA-MEM against the gene catalogue, determining the number of reads mapped to each gene sequence and normalising as described[29]. Functional annotation to KEGG pathways was carried out by mapping centroid sequences to the eggNOG-Mapper database[30] (version 4.5, downloaded on 1 March 2018) using Diamond software on our in-house server.

Microbial gene richness (MGR) was determined as described previously[29,31]. Briefly, data were downsized to adjust for sequencing depth and technical variability by randomly selecting 7 million reads mapped to the gene catalogue (of 11,005,136 genes) for each sample and then computing the mean number of genes drawn over 30 random samplings.

### Quantification and statistical analysis

*Pre-processing*. To correct for dilution differences between samples normalisation procedures were applied. Global metabolite ($^1$H-NMR spectra) data sets were corrected for dilution effects using median fold change normalisation[16]. Scaling to unit variance was then applied to serum and urine data sets, while pareto scaling was used for faecal datasets, due to the presence of dominant and variable oligosaccharide resonances. Targeted metabolites measured within urine samples were corrected for dilutional differences using osmolality and creatinine measurements[32]. Metagenome data were expressed as relative abundance. Taxa with low abundance (present in <30% of both subgroups) were excluded from downstream statistical analyses.

*Univariate analysis*. Due to the non-parametric nature of the results, differences between paired samples pre/post intervention in clinical data, quantified metabolites and within the gut microbiota were assessed for significance using the Wilcoxon Rank test (two-sided). Differences in non-paired data were assessed using the Mann–Whitney U test (two-sided). P-values were adjusted for multiple testing using the Benjamini-Hochberg (BH) False Discovery Rate method (pFDR). Phylogenetic Trees were generated to illustrate significant gut microbiota changes using the GraPhlAn[33] script in Python.

*Multivariate analysis*. Multivariate statistical analysis of normalised $^1$H-NMR spectra was performed using SIMCA 15 (Umetrics)[16]. Principal component analysis (PCA) was used to provide an overview of the data. Orthogonal Partial Least Squares

—Discriminant Analysis (OPLS-DA) models were established based on one predictive component and one orthogonal component to discriminate between samples from participants with and without T2D. Unit variance scaling was applied to the $^1$H-NMR spectral data. The fit and predictivity of the models obtained were determined by the $R^2X$ and $Q^2Y$ values respectively. Significant metabolites differentiating between groups were obtained from $^1$H-NMR OPLS-DA models after investigating $^1$H-NMR signals with correlation coefficient values higher than 0.35. Jack-knifed 95% confidence intervals of the coefficients were used to confirm statistical significance of the variables.

Paired global metabolic data, pre- and post-intervention, were analysed using Repeated Measures, Monte-Carlo Cross-Validation, PLS-DA (RM-MCCV-PLSDA)[34,35] using covariate adjusted projection to latent structures in MATLAB. Data were centred and scaled to account for the repeated-measures design. 1000 MCCV models were generated and used to calculate the mean cross-validated predictive component score ($T_{pred}$) and variance for each sample[35]. The fit and predictivity of the models obtained was determined by the $R^2X$ and $Q^2Y$ values respectively. Gaussian kernel density estimates of the $T_{pred}$ in each group were generated for visual interpretation[35]. A total of 25 bootstrap resamplings in each of the 1000 models was used to estimate the variance and mean coefficient for each variable and derive a $p$ value for each variable accordingly[35]. Benjamini-Hochberg false discovery corrections were performed and a variable was considered significant with a false discovery rate value (q) ≤ 0.01. Manhattan plots showing -$\log_{10}$(q) x sign of the variable regression coefficient for each variable within each RM-MCCV-PLSDA model were generated, with dotted lines added to illustrate the q value significance cut off level on the $\log_{10}$ scale.

Exploration of gut microbiota taxa was performed using Principal coordinates analysis (PCoA) of Bray-Curtis dissimilarity matrices (β-diversity) using the Vegan[36] function in R. Significance of group separation in β-diversity was assessed by permutational multivariate analysis of variance (PERMANOVA). Nested PERMANOVA was used for paired analyses pre- and post-intervention to account for the repeated measures design.

*Metabolite identification*. A combination of data-driven strategies such as such as SubseT Optimization by Reference Matching (STORM)[37] and Statistical TOtal Correlation SpectroscopY (STOCSY)[38] and analytical identification strategies were used to aid structural identification of significant discriminatory metabolites. Specifically, a catalogue of 1D $^1$H-NMR sequence with water pre-saturation and 2D NMR experiments such as J-Resolved spectroscopy, $^1$H-$^1$H TOtal Correlation SpectroscopY (TOCSY), $^1$H-$^1$H COrrelation SpectroscopY (COSY), $^1$H-$^{13}$C Hetero-nuclear Single Quantum Coherence (HSQC) and $^1$H-$^{13}$C Hetero-nuclear Multiple-Bond Correlation (HMBC) spectroscopy were performed. Finally, where possible, metabolites were confirmed by in situ spiking experiments using authentic chemical standards. See Supplementary Fig. 2 for an example $^1$H-NMR spectrum labelled with identified metabolites.

Relative concentrations of identified metabolites from $^1$H-NMR datasets were calculated from intensity measurements of a representative spectral peak of the metabolite, ensuring no overlap with signals from other metabolites.

*Euler diagram of metabolites*. A Euler diagram of identified metabolites from Serum, Urine and Faecal biofluids associated with Bariatric Surgery, Weight / BMI, T2D and Diet (pFDR <0.05) was generated using Eulerr in R (version 6.1.1)[39].

*DIABLO integration of omics datasets*. To probe relationships between data sets we used Data Integration Analysis for

**RYGB**  **VSG**

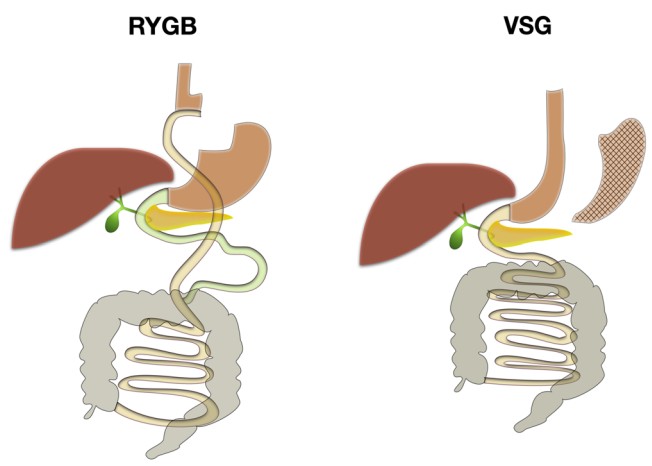

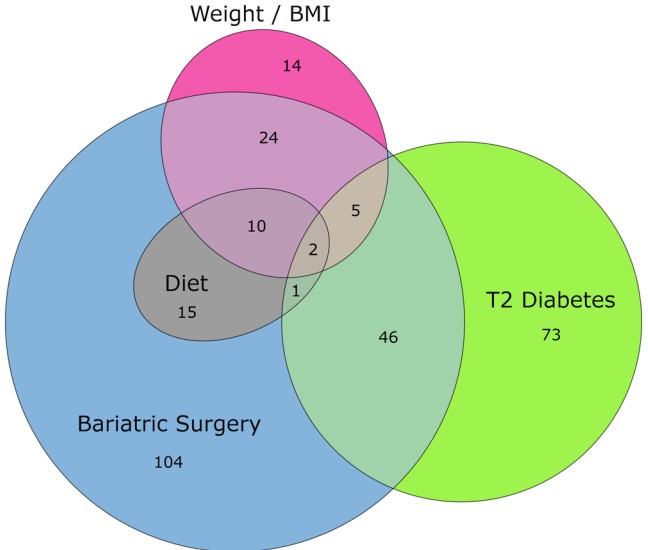

**Fig. 1 RYGB and VSG weight loss procedures.** Schematic of Roux-en-Y Gastric Bypass (RYGB) and Vertical Sleeve Gastrectomy (VSG) procedures, showing the respective anatomical changes.

**Fig. 2 Metabolites associated with bariatric surgery, body mass index and glycaemic control.** Euler diagram of identified metabolites associated with bariatric surgery, weight / body mass index (BMI), type 2 diabetes (T2D) and diet. Diagram shows identified metabolites from serum, urine and faecal biofluids with associations (pFDR < 0.05) to (1) post- versus pre-bariatric surgery, (2) lower weight / BMI, (3) lower glycated haemoglobin (HbA1c) / non-T2D Vs T2D, (4) lower dietary substrate / higher Alternative Healthy Eating Index (AHEI-2010) score. Metabolites with concordant changes are grouped together. Metabolites from each grouping and their associations are detailed in Supplementary Data 1.

Biomarker discovery using Latent cOmponents (DIABLO)[40], implemented through the mixOmics[41] package in R. DIABLO extends sparse generalized canonical correlation analysis (sGCCA)[42] to a classification framework. Resulting in a multi-omics integrative method that simultaneously identifies key variables correlated across different data types while discriminating between phenotypic groups.

Normalised datasets (gut microbiota species taxa, gut microbiota KEGG pathways (Levels 2-3) and quantified metabolites from urine, serum and faecal biofluids) were used with a full weighted design matrix where correlation was 0.1 between data matrices and 1 for the Y outcome, to result in a correlated and discriminant molecular signature[41]. To account for the repeated measures (pre/post procedure) experimental design, multilevel DIABLO (mDIABLO) models were constructed using within-subject variation matrices for each omics dataset[40,43,44]. Classification performance was assessed using the balanced error rate (BER) from cross-validation of samples, with BER = 0.5 x (false positive rate + false negative rate). BER scores range from 0-1, with a perfect classification model scoring 0, a random predictor 0.5 and a model with systematically incorrect predictions 1.

*Gut microbiota-metabolome associations.* Spearman's correlations between BMI, HbA1c and Microbiota-Metabolome datasets were generated in MATLAB. Partial Spearman's correlations were also performed to adjust for covariates. Corrected p values (pFDR) were used to select significant correlations. Significant (pFDR<0.01) first order correlations to BMI and HbA1c and cross-correlations between these variables were displayed using Cytoscape 3.8.0.[45] Correlations between gut microbiota, metabolite and dietary datasets were displayed using Complex-Heatmap in R[46]. Correlations with a pFDR value <0.05 are displayed, correlations with a pFDR <0.01 are highlighted. Hierarchical clustering of correlations was performed using Euclidean distances.

**Reporting summary**. Further information on research design is available in the Nature Research Reporting Summary linked to this article.

## Results
Serum samples were collected from 156 participants with obesity. Sixty-six participants had T2D, 26 had Impaired Glucose Tolerance (IGT) and 64 were non-diabetic. Complete sample sets of serum, 24-hour urine and stool were collected from 80 of these individuals (42 T2D, 11 IGT, 27 non-diabetic).

Forty-nine patients underwent bariatric surgery (VSG = 26, RYGB = 23; Fig. 1) and gave serum samples pre and 3-months post-surgery (19 T2D, 6 IGT). Twenty-seven of these participants (VSG = 14, RYGB = 13) gave complete sample sets of serum, 24-hour urine and stool pre and 3-months post-surgery. More patients with diabetes underwent RYGB than VSG (11/23 vs 8/26). Otherwise, baseline demographics were not significantly different between procedures. Full demographics are detailed in the Supplementary Tables 1 & 2.

Microbial and metabolic profiling indicated systematic differences relating to obesity, T2D and bariatric surgery (both RYGB and VSG), with metabolic signatures identified across the three biofluids (urine, serum, faecal water). Each condition had a specific set of metabolic correlates, with some overlap between groups. We identified 207 metabolites associated with bariatric surgery, 54 (26%) of these metabolites were characteristic of improved glycaemic control, 41 (20%) associated with BMI reduction and 28 (14%) were associated with dietary changes (Fig. 2 and Supplementary Data 1). Consistent with the observation that the mechanism for T2D resolution following bariatric surgery is partially independent of weight-loss, of the 175 metabolites associated with either T2D or BMI only 7 overlapped (4% commonality).

**Gut microbial differences between participants with and without T2D.** Shotgun metagenomic profiling did not identify a difference in microbial gene richness or β-diversity of the GM derived from the distal colon between participants with or without T2D at baseline (Supplementary Figs. 10 and 11). However, compositional analysis of the GM demonstrated lower relative abundance of the genera *Escherichia* (*Proteobacteria*), *Peptostreptococcaceae unclassified* (*Firmicutes*) and *Barnesiella* (*Bacteroidetes*) in individuals with T2D relative to individuals

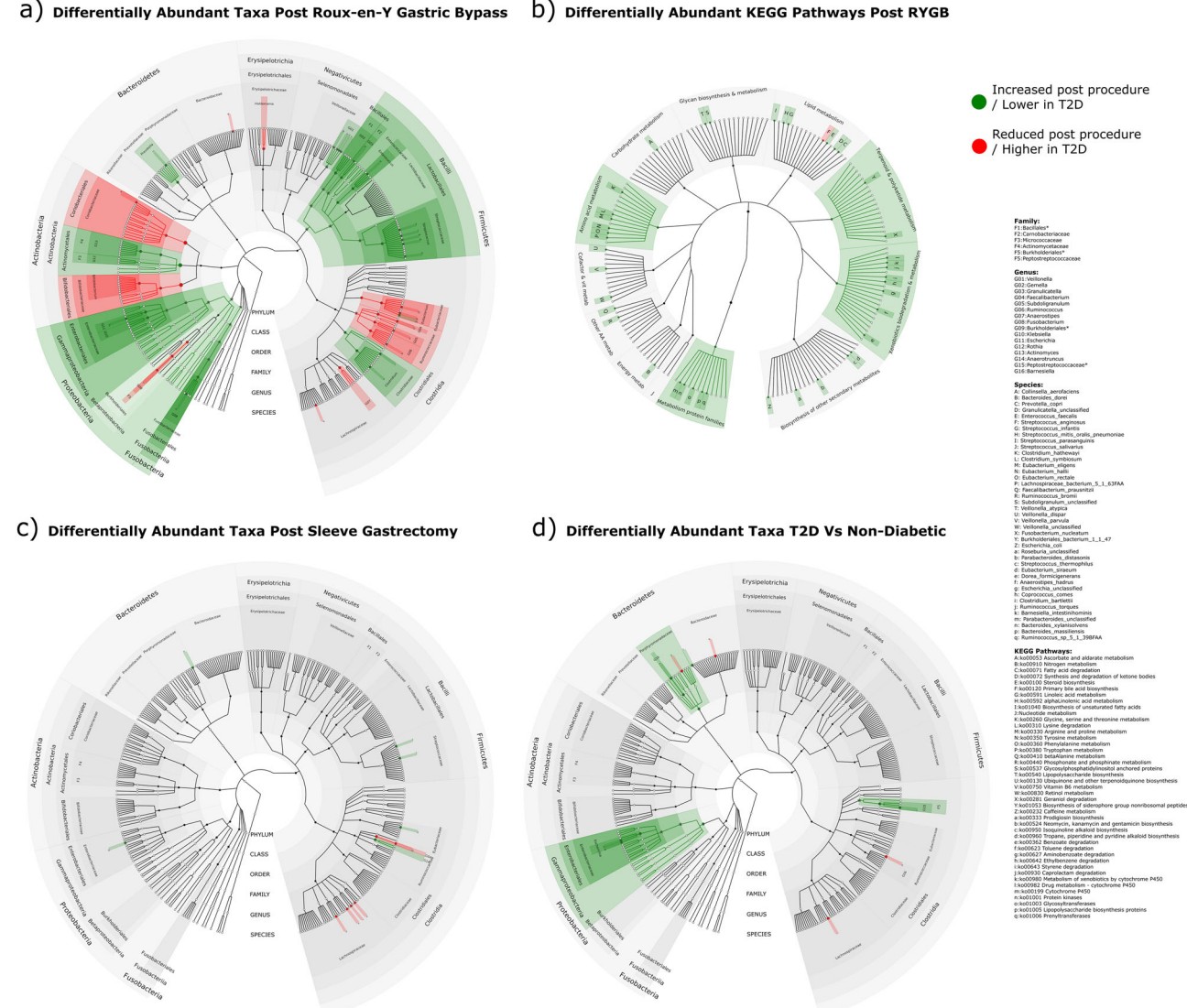

**Fig. 3 Gut microbiota changes after bariatric surgery and in participants ± T2D. a** Phylogenetic tree of significant differentially abundant taxa from phyla to species 3-months post Roux-en-Y Gastric Bypass (RYGB) (n = 13). **b** Phylogenetic tree of significant differentially abundant level 2 and level 3 Kyoto Encyclopaedia of Genes and Genomes (KEGG) pathways 3-months post RYGB (n = 13). **c** Phylogenetic tree of significant differentially abundant taxa from phyla to species 3-months post Vertical Sleeve Gastrectomy (VSG) (n = 14). **d** Phylogenetic tree of significant differentially abundant taxa from phyla to species between participants with type 2 diabetes (T2D) (n = 42) vs participants without diabetes (n = 27). Taxa / KEGG pathways significantly (p < 0.05) increased post-surgery (**a–c**) or lower in participants with T2D vs participants without diabetes (**d**) are shown in green, taxa / KEGG pathways significantly (p < 0.05) decreased post-surgery (**a–c**) or higher in participants with T2D vs participants without diabetes (**d**) are shown in red. Changes that remain significant after Benjamini-Hochberg multiple testing corrections (pFDR < 0.05) are denoted with an asterisk (*). Plots (**a–d**) are shown individually in Supplementary Figs. 12, 14, 15 and 17.

without diabetes. Individual species from the genera *Ruminococcus*, *Parabacteroides* and *Bacteroides* had higher relative abundance (Fig. 3d). Functional analysis of the GM found lower levels of KEGG pathways relating to cofactor and vitamin metabolism, including nicotinate and nicotinamide metabolism and one-carbon metabolism by folate in participants with T2D relative to participants without diabetes. Methane metabolism, streptomycin and neomycin biosynthesis, polycyclic aromatic hydrocarbon degradation and *D*-alanine metabolism pathways were also less prevalent in T2D, while bisphenol degradation pathways were more abundant.

**Metabolic differences between participants with and without T2D.** Individuals with T2D were metabolically distinct from BMI matched controls without T2D, as determined from both targeted

MS assays and global ¹H-NMR profiles. In participants with T2D, the serum secondary to primary (2:1) BA ratio was higher compared to participants without T2D at baseline. Conjugation of the primary BA cholic acid (CA), the overall glycine:taurine conjugation ratio of primary BAs and lithocholic acid were also higher in participants with T2D. Conversely, CA and the CA:CDCA (chenodeoxycholic acid) ratio were lower. In addition, conjugation of secondary BAs including tauro-ursodeoxycholic acid (TUDCA), taurohyocholic acid (THCA), glycohyocholic acid (GHCA) and the conjugated UDCA:UDCA ratio was lower. In faeces, 5α-cholanic acid-3α-ol-6-one was lower in individuals with T2D relative to control participants without T2D. Targeted GC-MS analysis found higher 2-hydroxybutyrate and lactate in urine and serum from participants with T2D relative to controls. Serum 2-methylbutyrate and isovalerate were also higher in

participants with T2D, while urinary butyrate was lower. Further quantitative analyses of serum metabolites found higher branched-chain AAs (BCAAs) leucine, isoleucine and valine, aromatic AAs (AAAs) phenylalanine and tyrosine, as well as alanine, methionine, glutamate, lysine and proline in volunteers with T2D relative to controls without diabetes. The biogenic amines 2-aminoadipate, methionine sulfoxide and sarcosine and short-chain acylcarnitines (C2-5, 9) were also present in higher concentrations in the group with T2D. Conversely, longer-chain acylcarnitines (C14, 16, 18) and a number of lyso-phosphatidylcholines, acyl-alkyl-phosphatidylcholines, longer-chain diacyl-phosphatidylcholines and sphingomyelins were present in lower concentrations in participants with T2D.

Orthogonal Partial Least Squares—Discriminant Analysis (OPLS-DA) models of global $^1$H-NMR spectra comparing individuals with and without T2D were generated. Serum and faeces produced robust models ($R^2Y = 0.61$, $Q^2Y = 0.44$ and $R^2Y = 0.59$, $Q^2Y = 0.35$, respectively) while urine produced the least robust model ($R^2Y = 0.80$, $Q^2Y = 0.12$, Supplementary Fig. 3). In serum, participants with T2D were characterised by higher concentrations of VLDL/LDL lipoproteins, BCAAs, lactate, alanine, proline, pyruvate, tyrosine and α-glucose relative to controls without T2D, whereas HDL, glutamine, glycerophosphocholine, phosphocholine/choline and histidine were lower. In faeces, individuals with T2D had higher levels of glycine and the anti-hyperglycaemia drug dimethylbiguanide (Metformin). Lactate, uracil, BCAAs, and tyrosine were lower relative to controls without diabetes. As expected, in urine, α- and β-glucose were higher in participants with T2D, while isobutyrate, glycine, creatine, creatinine, O-acetylcarnitine, N-methyl-2-pyridone-5-carboxamide, methylnicotinamide and formate were lower relative to participants without T2D.

Full details of metabolic differences in participants with T2D relative to participants without diabetes can be found in Supplementary Tables 3–5 and Supplementary Data 2–5.

**Integrative analysis of metabolic and gut microbiota profiles in individuals with and without T2D.** Multi-omic signatures of participants with T2D versus individuals without diabetes were modelled using Data Integration Analysis for Biomarker discovery using Latent cOmponents (DIABLO), with a cross-validated balanced error rate (BER) of 0.18, indicating good class separation at the systems level (see Supplementary Fig. 18). Volunteers with T2D were characterised by higher levels of lactate, glucose and alanine in serum; lactate, glucose and 2-hydroxybutyrate in urine and the presence of dimethylbiguanide (Metformin) in faeces compared to controls without diabetes. Whereas lower serum phosphatidylcholines and HDL lipoprotein; lower urinary glycine, trimethylamine and isobutyrate and faecal valine and uracil levels were characteristic of participants with T2D relative to participants without diabetes. Lower levels of GM from the *Peptostreptococcaceae unclassified* (*Firmicutes*) and *Barnesiella* (*Bacteroidetes*) genera were also seen in the T2D signature, as were lower levels of GM KEGG pathways including lipid and *N*-glycan biosynthesis; nicotinate, nicotinamide, methane, alanine and one-carbon metabolism; and biosynthesis of secondary metabolites such as streptomycin and neomycin.

**Clinical findings after bariatric surgery.** Significant weight-loss was achieved after both procedures, although the percent weight-loss was greater after RYGB ($p = 0.023$) (Fig. 4a). At 3-months post-surgery patients with T2D who underwent VSG and RYGB had mean (±standard deviation) glycated haemoglobin (HbA1c) reductions of 17.8 mmol/mol (±11.2) and 19.4 mmol/mol (±12.5)

respectively. Relative to baseline, VSG and RYGB patients had reductions in HbA1c of 27.6% (±12.1) and 28·6% (±14.2), respectively. Three months after VSG, 4/8 participants with T2D had complete diabetes resolution (HbA1c < 42 mmol/mol), 2/8 had partial resolution (HbA1c < 48 mmol/mol) and 2/8 had ongoing T2D (HbA1c ≥ 48 mmol/mol). Of 4 participants with IGT that underwent VSG, two had resolution. Three months after RYGB, 8/11 participants with T2D had complete diabetes resolution and 3/11 had ongoing T2D. Two participants with IGT underwent RYGB, both had complete resolution.

Three months following surgery there were large changes in dietary intake compared to baseline. As measured by three self-reported 24-h dietary recall questionnaires collected at each time point, there were significant reported reductions in calorie, carbohydrate, fat, fibre and sodium intake after both operations. In addition, protein and vegetable intake was significantly reduced after VSG, while saturated fat and sugar intake were significantly reduced after RYGB (Fig. 4c). Dietary healthiness, measured using the Alternative Healthy Eating Index (AHEI-2010)[13], was increased 3-months after RYGB and VSG surgery (median 42 vs 54, $p = 0.002$, Fig. 4b).

**Gut microbial changes after bariatric surgery.** Multivariate analysis demonstrated a change in β-diversity (Bray-Curtis dissimilarity) after RYGB (PERMANOVA $p = 0.002$), but not VSG (Supplementary Figures 13 and 16). There was no difference in microbial gene richness after either bariatric procedure. Compositional analysis revealed a major disruption to the GM after RYGB, but more subtle changes after VSG. Three months after RYGB, participants had: (i) Increased relative abundance of *Veillonella*, *Gemella*, *Granulicatella*, *Enterococcus*, *Streptococcus* and *Clostridium* (*Firmicutes*), *Fusobacterium* (*Fusobacteria*), *Klebsiella* and *Escherichia* (*Proteobacteria*), *Actinomyces* and *Anaerotruncus* (*Actinobacteria*) and *Prevotella* (*Bacteroidetes*); (ii) decreased relative abundance of *Holdemania*, *Eubacterium*, *Faecalibacterium*, *Subdoligranulum*, *Ruminococcus*, *Dorea* and *Anaerostipes* (*Firmicutes*), *Burkholderiales* (*Proteobacteria*), *Bifidobacterium* and *Collinsella* (*Actinobacteria*). Changes at each taxonomic rank from phylum to species are shown in Fig. 3a. Functional analysis found an increase in bacterial KEGG pathways pertaining to: AA metabolism, lipid metabolism including fatty acid degradation, α-linolenic acid metabolism; and xenobiotic biodegradation including benzoate, aminobenzoate and ethylbenzene degradation. Bile salt hydrolase (choloylglycine hydrolase) pathways were reduced ($p = 0.03$) (Fig. 3b).

Three months after VSG, participants had increased relative abundance of a select number of species within the genera *Streptococcus*, *Eubacterium* and *Anaerotruncus* (*Firmicutes*) and *Escherichia* (*Proteobacteria*) and decreased species within *Faecalibacterium*, *Dorea*, *Anaerostipes, Roseburia* and *Coprococcus* (*Firmicutes*) (Fig. 3c). Limited changes to KEGG pathways were found after VSG.

**Metabolic changes after bariatric surgery.** Increases in glycine conjugation relative to taurine in both primary and secondary serum BAs occurred after RYGB. Whereas decreased primary and conjugated primary BAs were noted after VSG. Additionally, glycohyocholic acid was increased after RYGB, while murocholic acid and isolithocholic acid were increased after VSG. After RYGB and VSG, both groups had increased secondary BA ursodeoxycholic acid (UDCA) and conjugated secondary BAs (GUDCA + TUDCA) in serum, likely to be predominantly due to the exogenous administration of UDCA for gallstone prophylaxis postoperatively in patients with a gallbladder in situ. Faecal levels of primary, secondary, conjugated primary and secondary and overall faecal BAs were decreased

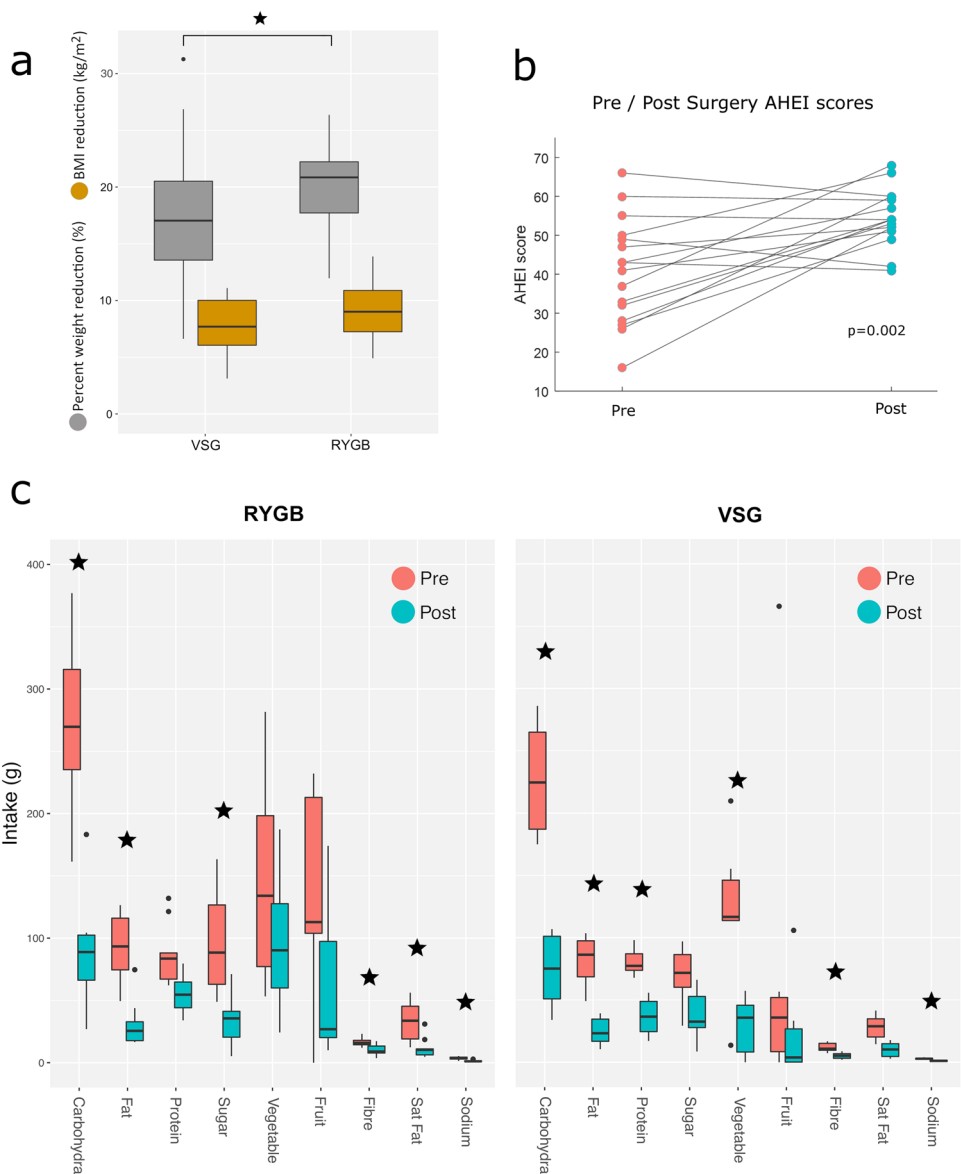

**Fig. 4 Clinical findings after bariatric surgery. a** Box plots of weight-loss 3-months post Roux-en-Y Gastric Bypass (RYGB) ($n = 23$) and Vertical Sleeve Gastrectomy (VSG) ($n = 26$) procedures. Patients who underwent VSG had a mean (SD) weight-loss, percent weight-loss and body mass index (BMI) reduction of 21.7 kg (6.8), 17.3% (5.6) and 7.9 kg/m² (2.2), respectively. Patients who underwent RYGB had a mean (SD) weight-loss, percent weight-loss and BMI reduction of 24.6 kg (7.0), 20.2% (4.1) and 9.3 kg/m² (2.4), respectively. Percent weight-loss was significantly greater in the RYGB group ($p = 0.02$). **b** Dietary healthiness measured by the Alternative Healthy Eating Index (AHEI) pre and 3-months post bariatric procedures. **c** Box plot of dietary intake pre and post RYGB & VSG procedures. Horizontal lines indicate the median, boxes indicate the interquartile range (IQR), whiskers extend to the upper adjacent value (Q3 + 1.5 x IQR) and lower adjacent value (Q1–1.5 x IQR), dots represent outliers. *$p < 0.05$.

after both procedures. Similar to serum there was an increase in the conversion ratio of primary to secondary BAs in faeces. Absolute 5α-cholanic acid-3α-ol-6-one and allolithocholic acid were increased after RYGB and VSG respectively. 5β-cholanic acid-3α-ol-12-one and 3,6/3,12-diketocholanic acid were reduced after RYGB. After RYGB, but not VSG, excretion of 2-methylbutyrate and isovalerate in urine and faeces increased. In serum, lactate was decreased after RYGB and showed a non-significant trend towards decreasing after VSG, consistent with the lower concentrations of serum lactate in participants without T2D compared to those with T2D. In faeces, acetate was reduced after both procedures, while butyrate and valerate reduced after VSG. Both procedures resulted in a decrease in the majority of serum AAs, including BCAAs and AAAs. However, glycine and serine were increased after both procedures, with glutamine also increased after RYGB. Serum kynurenine and sarcosine were

decreased and symmetric dimethylarginine was increased after both procedures. 2-aminoadipate was lower after VSG. Short-chain acyl-carnitines (C0, C3-5) and a large number of phosphatidylcholines and lyso-phosphatidylcholines species were reduced after both RYGB and VSG. Whereas, C2 and longer-chain acylcarnitines (C7, 8, 10, 14, 16, 18), sphingomyelins and predominantly longer-chain acyl-alkyl-phosphatidylcholines were increased, with similar findings after both interventions.

Repeated Measures, Monte-Carlo Cross-Validation, Partial Least Squares Discriminant Analysis (RM-MCCV-PLSDA) models of serum and urine global ¹H-NMR spectra found excellent separation between participants pre- and post-surgery, with robust models after both procedures (Fig. 5, Supplementary Figures 4-7). In serum, a number of significant changes were consistent after both RYGB and VSG, including altered AA

metabolism, matching results in the targeted AA analysis. Ketone bodies acetone, acetoacetate and 3-hydroxybutyrate were increased and evidence of changes to the tricarboxylic acid cycle was seen, with citrate increasing and pyruvate decreasing after surgery. VLDL/LDL, lipid glycerol and choline decreased 3-months after RYGB and VSG. The improvement in lipid profile was seen despite near universal use of statins preoperatively. In addition, lactate decreased after RYGB, consistent with the quantitative GC-MS analysis. In urine, a number of significant metabolic changes were also consistent after both RYGB and VSG procedures, including increased bacterially derived metabolites such as phenylacetylglutamine (PAG), *N*-methyl-4-pyridone-3-carboxamide, 4-cresylsulfate, hippurate, trimethylamine-*N*-oxide (TMAO) and 2-aminobutyrate. Decreased urinary excretion of lactate, 3-hydroxyisovalerate and AAs occurred and a reduction in analgaesic use was observed. RM-MCCV-PLSDA models of faecal samples analysed by $^1$H-NMR pre- and post-surgery produced a robust model for VSG but not RYGB (Supplementary Figures 8 & 9). Tyramine and β-alanine were increased after VSG. In keeping with the quantitative SCFA analysis, acetate, butyrate, valerate were decreased, as were isovalerate, lactate, methanol, formate, trimethylamine and phenylacetate.

Full details of metabolic changes after bariatric surgery can be found in Supplementary Tables 6–8 and Supplementary Data 6–9.

**Integrative analysis of metabolic and gut microbiota profiles after bariatric surgery**. Multi-omic signatures of the response to RYGB and VSG were modelled using multilevel DIABLO, with cross-validated balanced error rates (BER) of 0.021 and 0.085 respectively, indicating significant differences at the systems level in response to RYGB and VSG and excellent classification between pre- and post-surgery states (Fig. 6a–f). Each procedure had a distinct signature. Notably, RYGB was characterised by an increase in tyrosine and phenylalanine metabolism and benzoate and fatty acid degradation pathways in gut bacteria. This difference corresponded with increases in urinary metabolites of bacterial origin including PAG, hippurate and TMAO. In addition, there was increased serum glycine and glycine conjugation of CA and decreased tryptophan and valerate. A number of BA changes were seen in faeces including increased glyco-ursocholanic acid and 5β-cholanic acid 3α-ol-6-one. VSG was characterised by a decrease in a number of GM KEGG pathways. These correlated with decreased faecal SCFA levels, specifically acetate, butyrate and valerate. However, similar to RYGB a number of bacterially derived compounds were increased in the urine including indoxylsulfate, 4-cresylsulfate, hippurate and PAG. Serum changes consisted predominantly of decreased phosphatidylcholines.

A multi-omic signature differentiating between the two bariatric procedures was characterised by a greater increase in urinary compounds of microbial origin including PAG, indoxylsulfate, TMAO and 4-hydroxybutyrate after RYGB relative to VSG. The reduction in faecal acetate, butyrate, valerate was greater after VSG, while an increase in 2-methylbutyrate and isovalerate was specific to RYGB. Glycine conjugation of serum BAs was greater after RYGB. A number of bacterial species increased further after RYGB, including *Escherichia coli* and unclassified *Granulicatella* and *Gemella* species, as did GM KEGG pathways including tryptophan metabolism, benzoate and toluene degradation and biosynthesis of unsaturated fatty acids (Supplementary Fig. 19).

**Integrative correlation analyses of BMI and HbA1c**. BMI is closely associated with glycaemic control (HbA1c) and both decreased following bariatric surgery. However, interestingly,

BMI and HbA1c correlation networks had contrasting compositions (Fig. 7). BMI was strongly correlated to a number of bacterially derived factors. Species including *Escherichia coli*, *Streptococcus anginosus*, *Streptococcus parasanguinis*, *Clostridium hathewayi* and multiple *Veillonella* species were correlated with a lean phenotype, while *Eubacterium rectale* was correlated with increased adiposity. Subsequently a number of GM KEGG pathways such as tryptophan metabolism, linoleic acid metabolism, cytochrome P450, steroid biosynthesis and xenobiotic degradation were also correlated with lower BMI. Bacterially derived urinary metabolites PAG, 4-cresylsulfate, indoxylsulfate and 4-hydroxyphenylacetate were negatively correlated with BMI. While SCFAs acetate and valerate in faeces and isobutyrate in urine correlated positively, as did a number of BA species. Correlations to BMI were similar after correcting for glycaemic control (HbA1c) using partial correlations (Fig. 7c).

Conversely, HbA1c correlated positively with a range of serum AAs including BCAAs, acylcarnitines (C3, C5, C5-OH), lactate, kynurenine and 2-aminoadipate. In validation of the model, HbA1c increased with higher serum and urinary glucose, faecal metformin levels and age, as expected. Whereas serum HDL, sphingomyelins, glycerophophocholine and GHCA, urinary 2-aminobutyrate and glycine and faecal 5α-cholanic acid 3α-ol-6-one levels were negatively associated with HbA1c. To further isolate weight-independent correlates to HbA1c we conducted partial correlation analysis correcting for BMI and weight (Fig. 7d). Similarly, serum and urinary glucose, serum BCAAs, lipids and pyruvate correlated positively with HbA1c. Whereas serum glutamine, glycerophosphocholine, phosphocholine and HDL and urinary glycine correlated negatively.

**Influence of diet on gut microbial and metabolic profiles**. Metabolic changes associated with diet overlapped principally with the effects of bariatric surgery and BMI, but not T2D (Fig. 2). Carbohydrate and calorie intake correlated positively with a number of serum glycerophospholipids and sphingomyelins as well as phenylalanine, hydroxypropionylcarnitine (C3-OH), propenoylcarnitine (C3:1) and faecal acetate levels (Fig. 8). While urinary PAG, 4-cresylsulfate and serum acetate were negatively correlated with carbohydrates and calorie intake. PAG was also negatively correlated with fat intake, including trans and polyunsaturated fats. Serum glutamate was higher with higher levels of trans fat and tauro-ursodeoxycholic acid (TUDCA) in serum was negatively correlated with salt (sodium) intake. Fibre intake was positively correlated with serum proline and C24:1-OH sphingomyelin levels. While serum leucine levels were lower as diet healthiness (AHEI-2010 score) increased. Limited changes to the GM relating to diet were identified, but percentage carbohydrate intake correlated negatively with the *Proteobacteria* phylum.

**Discussion**
Bariatric surgery has been shown to reverse some of the deleterious effects to the multiple organ systems and metabolic pathways that are disrupted in T2D. However, the mechanisms behind the resolution of T2D are not well understood. It is thought that a complex interplay of weight dependent and independent factors[47] are involved including: reduced adipose tissue leading to reduced inflammation and improved insulin sensitivity[48]; changes in the architecture of pancreatic islets such as increased beta cell mass[49,50]; changes in energy homoeostasis and mitochondrial function[51]; restoration of bile acid levels and their impact on the farnesoid-X receptor (FXR) and the transmembrane G protein-coupled receptor 5 (TGR5)[52,53]; enhanced release of gut hormones (e.g. GLP1[54], ghrelin[55]); reversal of

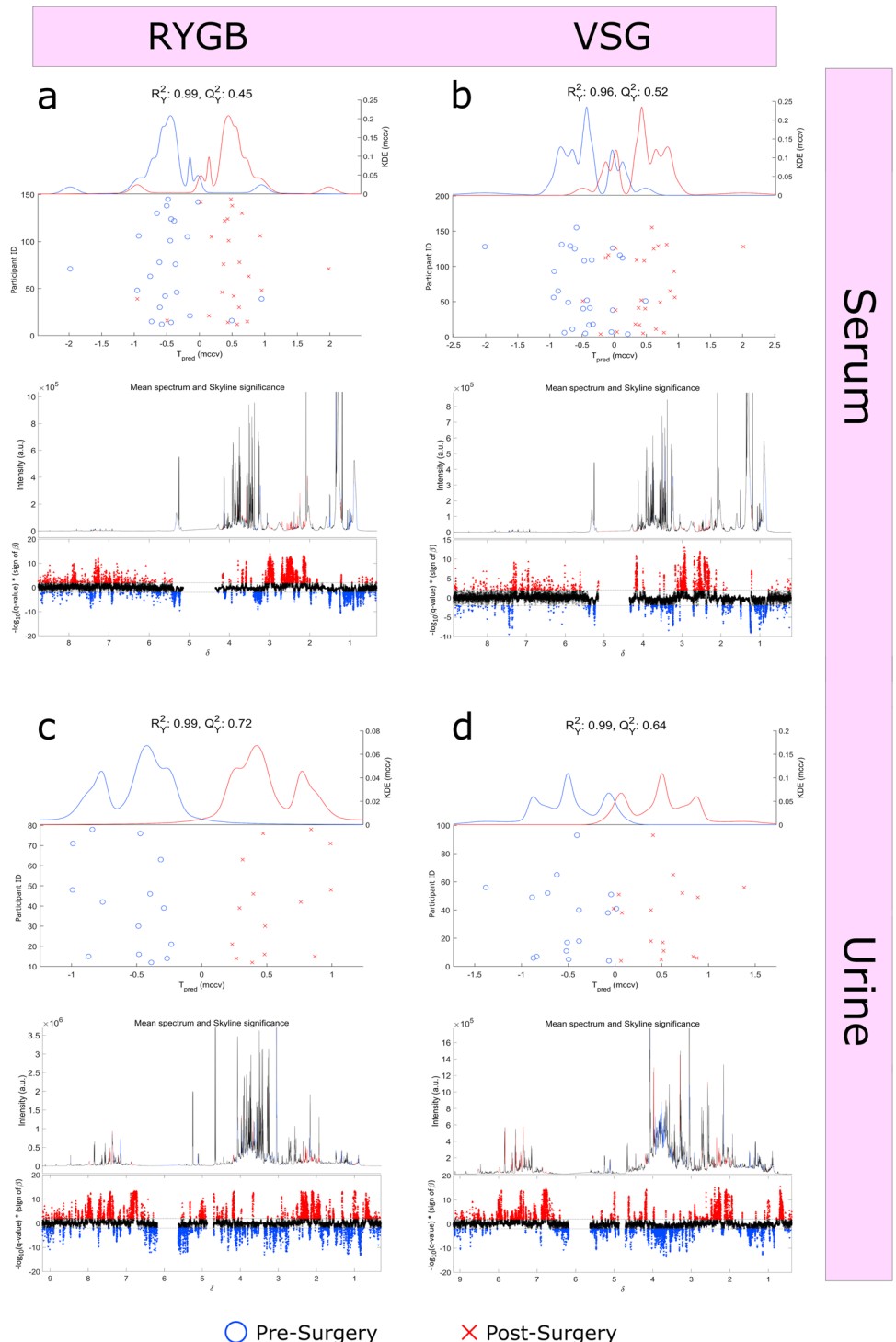

**Fig. 5 Metabolite changes after RYGB and VSG procedures, assessed by ¹H-NMR spectroscopy.** Repeated Measures, Monte-Carlo Cross-Validation, Partial Least Squares Discriminant Analysis (RM-MCCV-PLSDA) models were generated using ¹H – Nuclear Magnetic Resonance (¹H-NMR) spectra derived from serum and urine biofluids pre and 3-months post Roux-en-Y Gastric Bypass (RYGB) and Vertical Sleeve Gastrectomy (VSG) procedures. Model scores: (**a**) Serum RYGB ($n = 23$, R²Y 0.99, Q²Y 0.45), (**b**) Serum VSG ($n = 26$, R²Y 0.96, Q²Y 0.52), (**c**) Urine RYGB ($n = 14$, R²Y 0.99, Q²Y 0.72), (**d**) Urine VSG ($n = 16$, R²Y 0.99, Q²Y 0.64), (Supplementary Fig. 9a) Faeces RYGB ($n = 10$, R²Y 0.94, Q²Y 0.06), (Supplementary Fig. 9b) Faeces VSG ($n = 14$, R²Y 0.98, Q²Y 0.66). Upper panels: RM-MCCV-PLSDA scores plots comparing participant samples pre and 3-months post bariatric surgery. Models are comprised of 1 predictive and 1 orthogonal component. Lower panels: Mean ¹H-NMR spectrum and Manhattan plot. Manhattan plot showing -log₁₀(pFDR) x sign of the variable regression coefficient (β) for each variable within the RM-MCCV-PLSDA model. Dotted lines illustrate the pFDR significance cut off level (0.01) on the log₁₀ scale. Spectra considered significant are highlighted in the Manhattan plot and mean spectrum. Red metabolites (¹H-NMR signals) are significantly increased post-surgery, blue metabolites (¹H-NMR signals) are significantly decreased post-surgery.

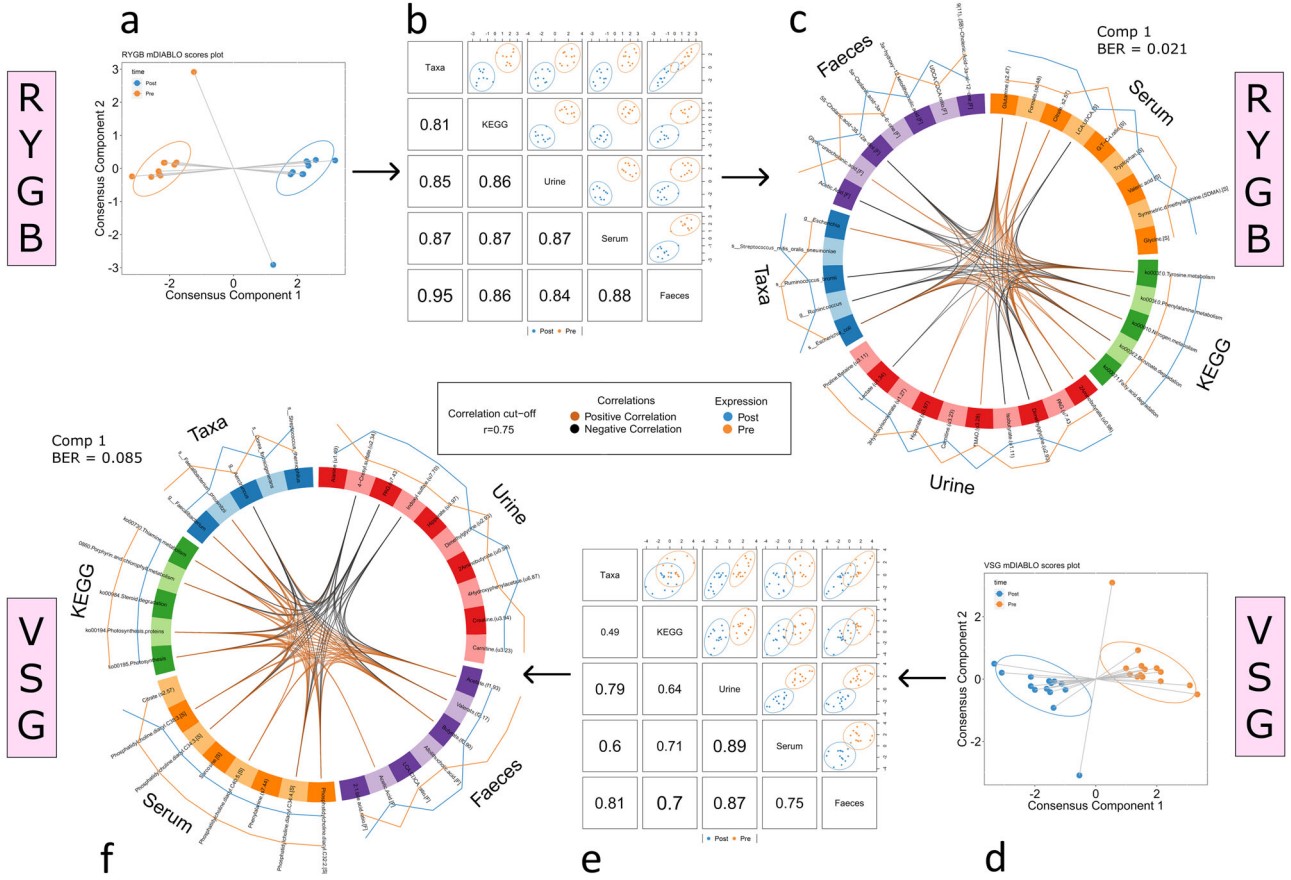

**Fig. 6 Integration of metabolite and gut microbiota datasets.** Multi-omic datasets pre and 3-months post Roux-en-Y Gastric Bypass (RYGB) (**a–c**) and Vertical Sleeve Gastrectomy (VSG) (**d–f**) procedures were integrated using multilevel Data Integration Analysis for Biomarker discovery using Latent cOmponents (mDIABLO). **a, d** Scores plots with samples projected in latent space for RYGB ($n = 10$) and VSG ($n = 14$) models respectively. Classes (pre-/post-surgery) are discriminated along component 1. **b, e** Scores plots derived from component 1 of individual datasets, showing correlations between variables from each dataset. **c, f** Variables from component 1 (coloured by dataset) discriminating between pre/post timepoints are displayed. Differences in expression of variables pre-/post-surgery are shown by the colour-coded outer lines. Correlations ($r > 0.75$) between variables are shown by the colour-coded inner lines. Cross-validated balanced error rates (BER) for each model are shown. Ellipses correspond to 95% confidence intervals.

reduced enteroplasticity[56]; and via modulation of the gut microbiome, which is known to contribute to BMI and other risk factors for T2D[57]. In addition, bariatric surgery has been shown to normalise gene expression that is dysregulated in T2D, particularly mitochondrial genes and those involved with aerobic metabolism[58]. Here we show the contrasting impact of T2D and bariatric surgery on multiple pathways such as aromatic amino acid metabolism (tyrosine), branched chain amino acid metabolism (valine, leucine, isoleucine), one-carbon metabolism (methionine, sarcosine), short chain acyl carnitines (C3, C4, C5:1) and anaerobic glycolysis (lactate, pyruvate), which were higher in T2D and reduced following bariatric surgery. Similarly, other amino acids (glutamine, histidine), bile acids (GHCA, THCA, GUDCA) and lipids (HDL, acyl-alkyl-phosphatidylcholines C32:1, C38:5, C40:5, C42:5, C44:5, C44:6, Sphingomyelins C16:0, C18:0, C18:1, C20:2, C24:1, C26:1) were lower in T2D and increased following bariatric surgery.

The associations of these metabolites with BMI and glycaemic control are described below. However, although the differential metabolite set representative of bariatric surgery overlapped with both diabetes (19.3% commonality) and BMI (18.6% commonality), the percentage overlap between diabetes and BMI was minimal (4.0% commonality). Thus, it appears that the change in metabolism with respect to resolution of T2D is at least in part independent of BMI reduction, consistent with the observation that BMI and HbA1c were differentially associated with metabolic profiles (Fig. 2 & 7).

As previously described[59] the clinical and metabolic impact of RYGB was greater than VSG. The RYGB procedure was strongly associated with altered functionality of the gut microbiome as reflected in the urine, serum and faecal metabolomes and in the KEGG pathways associated with the altered microbiome. The microbiome associated with T2D was distinct from the microbiome characteristic of participants without T2D, but in general the perturbation caused by T2D was of a lower magnitude than the changes in microbial structure and function observed post bariatric surgery. The lesser extent of GM differences in participants with and without T2D relative to the effects of bariatric surgery, coupled with the strong correlation of the GM to BMI but not HbA1c (Fig. 7), suggests that the GM predominantly effects weight-dependent mechanisms. However, although the impact of T2D on the GM was more subtle, nevertheless we identified changes in several metabolite groups that correlated with GM changes after bariatric surgery. As discussed below, some of these metabolites, such as BCAAs, influence glycaemic control and may be important in weight-independent mechanisms of T2D resolution following bariatric surgery.

**Multi-omic comparison of participants with T2D to BMI matched controls without diabetes.** Although we found differences in various individual taxa within the GM of participants with T2D compared to BMI matched controls without T2D, there was no

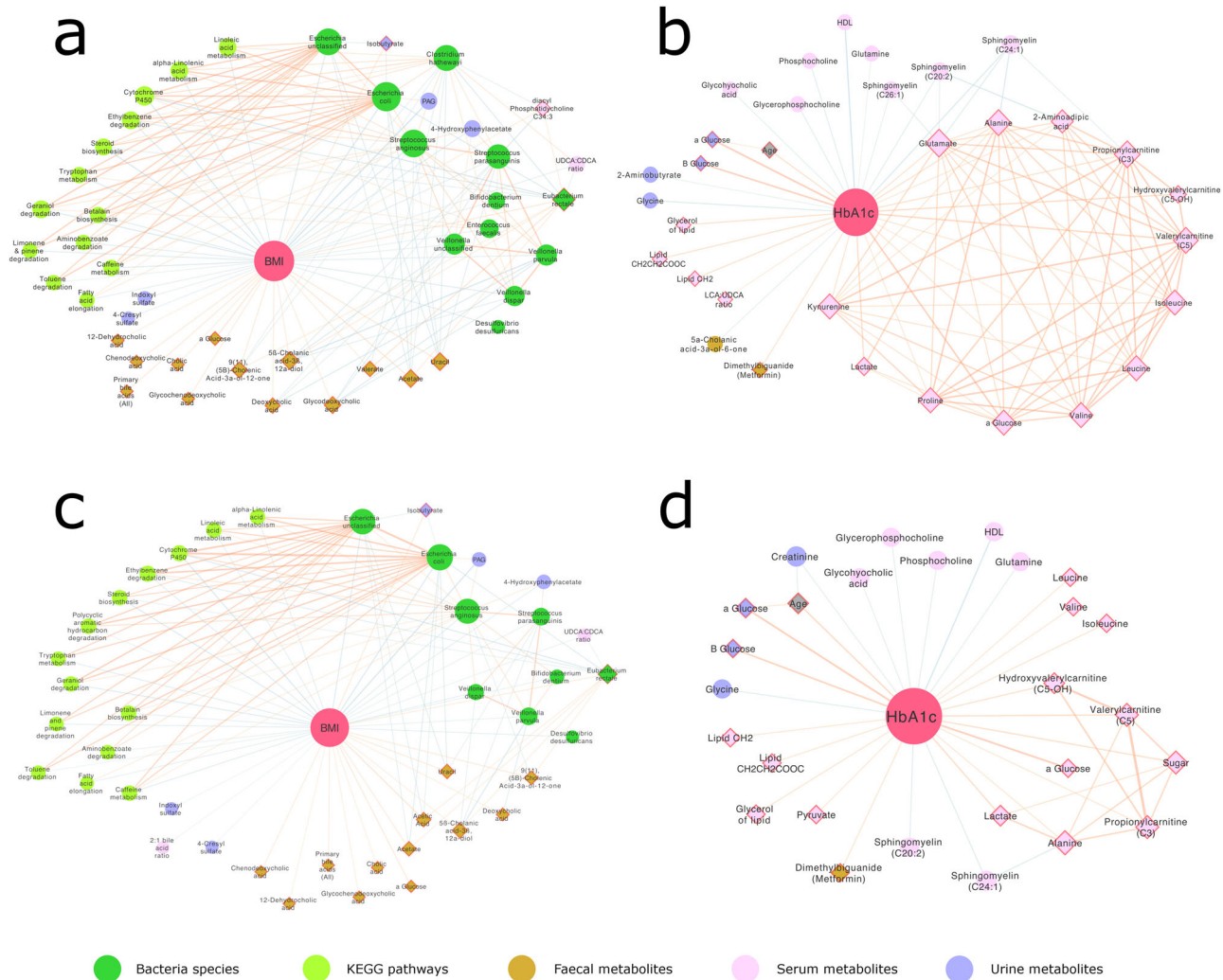

**Fig. 7 Metabolite and gut microbiota correlations with BMI and HbA1c.** First-order Spearman correlations to (**a**) body mass index (BMI) and (**b**) glycated haemoglobin (HbA1c) derived from metabolite and microbiota datasets. **c** Partial Spearman correlations to BMI, correcting for HbA1c. **d** Partial Spearman correlations to HbA1c, correcting for BMI and weight (weight independent HbA1c correlations) ($n = 108$). Correlations with a corrected pFDR < 0.01 are shown. Variables with a negative correlation to BMI / HbA1c are displayed within a circle, variables with a positive correlation have a diamond shape and red outline. Node size is proportional to the number of significant correlations to that variable. Positive correlations between variables are shown with orange lines, negative correlations have blue lines. Line thickness is proportional to the correlation strength (r).

overall difference detected in β-diversity. Similarly, others have found only moderate differences in individuals with T2D and large inter-individual variability[1,60]. However, in general the differences in microbial ecology between participants with and without T2D was consistent with the existing literature. Consistent with our observations in the current study, *Ruminococcus* and *Parabacteroides* have a higher relative abundance in T2D rodent models compared to animals without diabetes[61,62] and *Ruminococcus* has been shown to exert pro-inflammatory effects and also to promote pathogenesis in type 1 diabetes[63]. Similarly, lower relative concentrations of *Barnsiella* have been associated with T2D in rodent models[64]. In contrast, the role of Bacteroides is less clear with some studies reporting lower abundancy of *Bacteroides* in T2D[61]. Some of the differences in this T2D cohort, such as lower *Clostridium bartlettii* (aka *Intestinibacter bartlettii*) levels relative to controls without diabetes, are in keeping with changes due to metformin use (86% metformin use in T2D group, see Supplementary Table 1 & 2)[65,66]. However, other differences in participants with T2D such as lower *Escherichia coli* were apparent despite metformin's well characterised action in increasing its relative abundance[65,66]. *Escherichia* subsequently increased following both RYGB and VSG surgery.

In contrast, large metabolic dissimilarities occurred in participants with T2D compared to BMI matched controls without diabetes and we have defined a T2D metabolic signature, characterised by altered branched chain and aromatic AA, one-carbon, acylcarnitine, lipid, BA and SCFA metabolism. The association of lysine and 2-aminoadipate with HbA1c is consistent with the fact that both metabolites have been associated with increased risk of T2D in the PREDIMED and Framingham Offspring Studies[67,68]. However, other studies seem to indicate that 2-aminoadipate can modulate insulin secretion and reduce the impact of diabetes[69]. These contrasting results may be due to Maillard reactions which are increased in diabetes and other age related diseases[70]. Similarly, inflammation has been shown to promote the conversion of tryptophan to kynurenine and increased circulating kynurenine has been previously associated with HbA1c and diabetes[71,72]. Likewise, methionine is susceptible to oxidation to methionine sulfoxide, which was increased in serum in participants with T2D. This has been associated with ageing and is consistent with higher oxidative stress in individuals with T2D[73]. Metformin is known to exert an influence on the serum metabolite profile and is associated with

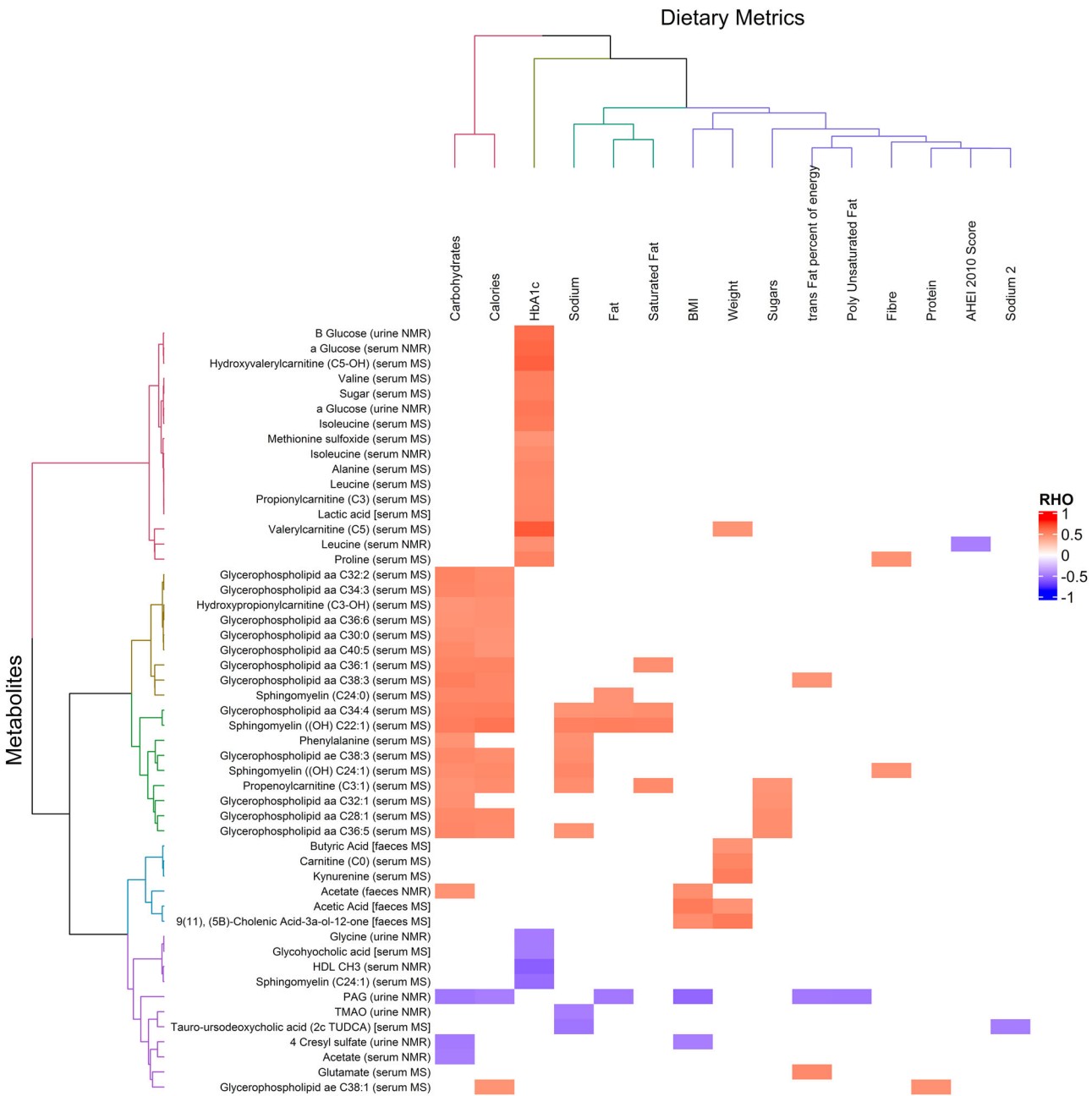

**Fig. 8 Metabolite—dietary intake correlations.** Significant Spearman's correlations (pFDR < 0.05) between measured metabolites and dietary intake are shown and shaded according to strength of correlation coefficient (Rho), ($n = 65$). The corresponding biofluid (urine, serum and faeces) of each measured metabolite is noted.

lower concentrations of lipoproteins, *N*-acetylated glycoproteins, lactate and glucose, along with increased relative concentrations of TMAO and 3-hydroxybutyrate[74]. Thus, metformin shifts the serum profile towards a non-diabetic state and most of the metabolic differences observed in the current study do not appear to be differentially modulated by Metformin.

**Multi-omic assessment of changes post RYGB and VSG surgery.** Reductions in the intake of all food groups, in particular carbohydrate, fat, sugar and fibre are commonly noted after RYGB and VSG. In addition to the restrictive elements of both procedures through a reduction in stomach size, bariatric procedures are known to reduce hunger and increase satiety through modulation of anorexigenic hormones[75]. Patients also have a lower brain-hedonic response to food[76] and altered food

preferences[77]. This change appears to be reflected here with an increase in dietary healthiness, measured using the AHEI-2010 score, after surgery. However, it should be noted that all self-reported dietary data are subject to misreporting[78]. Although both surgical procedures increased diet healthiness, several microbial and metabolic changes post-surgery were distinct between interventions thereby implying that dietary change was not the main driver in these effects. Nevertheless, a number of metabolic effects due to dietary changes were observed and are discussed in the relevant sections below.

The magnitude of alteration in faecal bacterial composition following bariatric surgery was far greater than the perturbation in the GM that was associated with T2D. This was also consistent with the alteration in microbial metabolites identified in the serum, faeces, and particularly the urine following bariatric

surgery (hippurate, PAG, 4-cresyl sulfate, indoxyl sulfate and TMAO) whereas these metabolites did not strongly differentiate individuals without diabetes from those with T2D. GM composition changes along the intestine's length due to a gradient in a number of factors such as nutrient availability, oxygen levels, pH and antimicrobial activity including BA levels[79]. The changes to the GM after RYGB in this cohort represent a shift in the colonic bacteria towards those usually found in higher concentrations in the small bowel[80,81]. Additionally, representation of a number of obligate anaerobes increased. Whereas fermentative bacteria, usually found in high concentrations in the colon, decreased after RYGB[81,82]. As expected, due to the fact that VSG maintains continuity of the gastrointestinal tract, the bacterial changes following VSG were more subtle. Nevertheless, several species increased after VSG from genera that increased after RYGB. Initial studies investigating obesity and the gut microbiota found a higher *Firmicutes/Bacteroidetes* ratio in individuals with obesity[2,83], while other studies have reported contrasting results, finding the opposite changes or no difference[84,85]. We found no significant change after either procedure, supporting the notion that the picture is more complex than changes at a phylum level and that changes at lower taxonomic levels and at a functional level may be more relevant[86]. Functional analysis of our cohort revealed a significant shift towards increased bacterial proteolytic fermentation (putrefaction) pathways after RYGB. This increase occurred despite a decrease in dietary protein consumption, although it is possible that malabsorption resulting from altered small-intestine anatomy leads to higher AA concentrations reaching the colon. The change was corroborated by an increase in a number of bacterially derived metabolites, generated through the fermentation of AAs. These included urinary PAG, indoxylsulfate and 4-cresylsulfate derived from the bacterial metabolism of phenylalanine, tryptophan and tyrosine respectively. Each of these amino acid – microbial metabolite pairs had a significant inverse correlation, consistent with upregulated fermentation of proteins and AAs. Isovalerate and 2-methylbutyrate, derived from bacterial-degradation of BCAAs, were also increased in faeces and urine after RYGB. Urinary PAG has previously been associated with a lean phenotype[87] and was found to be higher after RYGB in a rat model[88]. In this study, PAG was also correlated with a healthier diet (lower carbohydrate and fat intake, including trans and polyunsaturated fats). Similarly, the increase in hippurate, the glycine conjugate of benzoate, post-surgery is consistent with a lean phenotype and increased fruit intake, as well as reduced risk of metabolic syndrome independent of diet[89]. Although bacterial pathways relating to proteolytic fermentation were not significantly increased after VSG, a similar but less pronounced increase in related metabolites also occurred, suggesting that these pathways were also functionally increased after VSG.

Perturbations in AA homoeostasis, particularly BCAA and AAAs are associated with insulin resistance[90] and future risk of diabetes[91]. In this study we identified a higher Fischer ratio (BCAA/AAA), previously associated with worsening liver function, in participants with T2D relative to participants without diabetes[92,93]. BCAA and AAAs both reduced following surgery, but the Fischer ratio only reduced following surgery in participants with T2D and impaired glucose tolerance and was not significantly altered following surgery in individuals without diabetes. The association of BCAA with diabetes is thought to be causative[94]. BCAA infusions in rats and humans leads to the development of insulin resistance[95]. This led us to conclude that the observed changes in AA profiles after surgery are likely to be an important factor driving improved insulin resistance. Our findings suggest that the changes in AA profiles resulted in part

from increased AA putrefaction by the GM after RYGB. Consequently, RYGB may confer a metabolic advantage for patients with T2D compared to VSG, through decreased availability and absorption of BCAA from the gut. In support of this, germ-free mice have significantly altered profiles of AAs absorbed from the gut via the portal vein relative to those with a normal GM[96]. This difference is due to the large number of bacteria involved in AA biosynthesis and fermentation[97,98]. In a prior study, oral gavage of mice with *Bacteroides thetaiotaomicron* led to a reduction in glutamate, phenylalanine, leucine and valine[10]. Here two species of *Paraprevotella*, highly capable of producing AAs, correlated with serum alanine levels. Perhaps more importantly we also identified a number of negative correlations between bacterial species and serum AA levels, indicating that fermentation of AAs by gut bacteria is an important pathway influencing human AA profiles. In particular, species from the genera *Streptococcus* and *Clostridium*, important AA fermenters[98], correlated with a reduction in serum BCAAs (Supplementary Figure 20). However, further studies are needed that investigate the combined microbial, dietary and human contribution towards circulating levels of BCAAs. For example, we also note that the BCAA leucine was negatively correlated with dietary healthiness (AHEI-2010 score) in this cohort.

In keeping with greater changes to the GM following RYGB, urinary TMAO was significantly increased after RYGB but not VSG. Reduction of TMAO and other dietary nutrients such as choline and *L*-carnitine to trimethylamine (TMA) is performed by the GM (predominantly *Enterobacteriaceae*). Absorbed TMA is then converted back to TMAO by host hepatic enzymes[99]. TMAO has been described as pro-atherogenic and high serum levels of TMAO have been proposed to be a predictor of cardiovascular disease[91,100]. However, the causative effect of cardiovascular disease from high TMAO is disputed and these results may have been due to confounders such as reduced kidney function and poor metabolic control[101]. Other evidence such as the presence of high concentrations of TMAO in elite athletes, as well as in the urine of Japanese populations whose diet contains a high portion of fish and who have a low risk of cardiovascular disease, suggests that the causal role of TMAO in cardiovascular disease is complex and conditional on a wealth of host-microbiota factors[102,103]. Moreover, TMAO was recently found to protect against impaired glucose tolerance and reduce endoplasmic reticulum stress[104].

The initial gateway step in biotransformation of primary conjugated BAs to secondary BAs is performed by gut bacteria containing the bile salt hydrolase (BSH) enzyme. Interestingly, although we observed increased secondary BAs, we found that the BSH gene load was decreased after RYGB with no overall change after VSG. Furthermore, there were no significant changes in other BA enzymes such as hydroxysteroid dehydrogenases. We concluded that the profound changes in the BA pool composition observed after both procedures, including increased secondary BAs, result predominantly from changes in host factors such as altered hepatic processing and BA re-absorption. In addition, there was a consistent increase in glycine conjugation of BAs relative to taurine in serum after RYGB. Changes in glycine/taurine conjugation can result from changes in their bioavailability in the liver[105] and certainly there was increased serum glycine in this cohort. In addition, bacterial glycine metabolism pathways were increased after RYGB and this may also influence the bioavailability of glycine for conjugation. Indeed, germ-free animals excrete almost exclusively tauro-conjugated BAs[106]. Interestingly, GHCA had a strong negative correlation with HbA1c in this cohort. Diabetes has been associated with lower serum concentrations of HCA species and they are strong predictors of metabolic disease[107]. Administration of hyocholic

acid by others increased serum fasting GLP-1 in healthy and diabetic mouse models by simultaneously activating TGR5 and inhibiting FXR, a unique mechanism not found in other BA species[108]. However, the overall metabolic effects of differences in the BA pool and conjugation patterns are difficult to predict[53].

SCFAs are produced through the bacterial fermentation of dietary fibre and complex carbohydrates. Faecal acetate decreased after both procedures, likely due to a large reduction in dietary substrate. Indeed, faecal acetate was negatively correlated with dietary carbohydrate and calorie intake. However, the reduction in butyrate and valerate seen after VSG was not replicated after RYGB, despite a similar reduction in dietary substrate. This discrepancy suggests that the GM following RYGB is able to produce these SCFA more readily than the microbiota after VSG. The SCFA signature is important as each SCFA has a unique impact on the host. Acetate production leads to a positive feedback loop that increases appetite, induces lipid deposition in liver and skeletal muscle, and increases insulin resistance[109]. Whereas butyrate has a beneficial role in host satiety, insulin resistance and colonocyte health[110].

Crucially, although the overall changes after bariatric surgery appear to be towards a healthier phenotype there may be some negative consequences of surgery. Increased protein metabolism within the gut is usually considered to be harmful[111]. Putrefaction results in the production of toxic compounds such as amines, sulphides and ammonia[112], whilst phenols and indoles, increased after bariatric surgery in this cohort, are reported to be pro-inflammatory and cytotoxic[113,114].

In conclusion, metabolic changes post-surgery were achieved by both weight-dependent and weight-independent processes and we have identified multi-omic signatures specific to obese and T2D states at the systems level, some of which demonstrated contrasting patters when compared with the effects of bariatric surgery. BMI correlated inversely with bacterially derived urinary metabolites such as PAG, 4-cresylsulfate and indoxylsulfate and positively with faecal acetate and valerate, whereas HbA1c showed stronger correlation with serum AAs, acylcarnitines, kynurenine and 2-aminoadipate. Greater functional and taxonomic changes were observed in the GM following RYGB compared to VSG. These microbial changes, particularly after RYGB, appeared to influence the complex relationship between the GM and host metabolism. The abundance of amino acid metabolism pathways within the GM and corresponding metabolites of protein putrefaction increased after RYGB, despite reduced dietary protein intake, and correlated with decreased serum BCAAs.

Although this is one of the largest studies to date integrating metagenomic, metabolomic and dietary data from bariatric surgery patients and presents several novel avenues for future study, we acknowledge our approach is not without limitations. Specifically, metagenomics provides functional predictions with respect to microbial metabolite production. To allow us to fully elucidate the contribution of the microbiota to the metabolites we detect, a standardized dietary intervention incorporating one or more stable-isotope-labelled substrates would be required in our cohort. This approach would not only allow us to pinpoint those metabolites produced via microbial biotransformations but would also allow us to incorporate metabolic flux analysis into our metagenomic analyses. Ultimately, this and further mechanistic work is needed to better understand GM–host co-metabolism pathways and to establish their full effects on human health.

## Data availability

Metagenomic data has been deposited with GenBank, EMBL and DDBJ databases under the BioProject accession number PRJNA473348.

Further metagenomic and metabolic source data have been deposited at Mendeley Data: https://doi.org/10.17632/t76nm3yfzh.3[115].

All other data are available from the corresponding author on reasonable request.

## Code availability

The code to execute RM-MCCV-PLSDA (and also PLS, OSC-PLS, CA-PLS) is provided in https://bitbucket.org/jmp111/capls/src. The code for executing both the STOCSY and STORM algorithms is in https://bitbucket.org/jmp111/storm/src. These codes can be executed in a Matlab environment.

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

## Acknowledgements

This research was funded by the Diabetes Research and Wellness Foundation through the Sutherland-Earl Clinical Research Fellowship 2015 awarded to NP. The research was supported by the National Institute for Health Research (NIHR) Imperial Biomedical Research Centre (BRC) and Clinical Research Network (CRN). The views expressed are those of the authors and not necessarily those of the NHS, NIHR or UK Department of Health and Social Care. We are grateful to the Imperial Weight Centre for facilitating this research. This work used the computing resources of the UK MEDical BIOinformatics partnership—aggregation, integration, visualisation and analysis of large, complex data (UK Med-Bio)—which was supported by the Medical Research Council (grant number MR/L01632X/1). IGP is supported by a NIHR Career Development Research Fellowship (NIHR-CDF-2017-10-032). JMP is supported by a Rutherford Fund Fellowship at Health Data Research (HDR) UK (MR/S004033/1). GF is supported by an NIHR Senior Investigator award. LH was in receipt of an MRC Intermediate Research Fellowship in Data Science (MR/L01632X/1, UK MED-BIO). EH is supported by the Department of Jobs, Tourism, Science and Innovation, Government of Western Australian through the Premier's Science Fellowship Program and by the Australian Research Council Laureate Fellowship Scheme.

## Author contributions

Conceptualisation: N.P., E.H., J.M., A.D. Funding acquisition: N.P., S.P. Data curation: N.P., D.Y., B.G. Microbiome Investigation: N.P., L.H., J.M. Metabolite Investigation: N.P., I.G.P., E.H. Microbiome Analysis: L.H., N.P. Metabolite Analysis: N.P., I.G.P., E.H. Dietary Analysis: A.K., N.P., G.F. Software: J.M.P., G.F. Supervision: E.H., S.P., J.M., G.F., A.D., H.A. Resources: E.H., S.P., J.M., G.F., A.D. Writing—Original Draft: NP. Writing—Review & Editing: All authors.

## Competing interests

H.A. is Chief Scientific Officer at Preemptive Health and Medicine, Flagship Pioneering. A.D. is Executive Chair at Preemptive Health and Medicine, Flagship Pioneering. J.R.M. has received consultancy from EnteroBiotix Limited, Cultech Limited and Kineticos. I.G.P., E.H., G.F. are shareholders and directors of Melico Sciences Limited, J.M.P. is a shareholder in Melico Sciences Limited. All other authors have no competing interests to declare.
