## [Peer Review File · Communications Medicine]

Reviewers' comments:

Reviewer #1 (Remarks to the Author):

In this manuscript, Penney and colleagues assessed, using a multi-omics approach, the similarities and differences in host-microbe responses to bariatric surgery, glycaemic control and obesity. The authors report that bariatric surgery reversed a number of changes in the microbe that are characteristic for T2D.

The manuscript is well written and easy to understand. It deals with an interesting and important topic, the role of the host-microbe in the metabolic effects of bariatric surgery and its effects on improving energy and glucose metabolism.

A key weakness of the here presented work is that authors fail to provide any mechanistic evidence that links the reported host-microbe responses to the metabolic effects of bariatric surgery, glycaemic control and obesity. The results, despite interesting, are quite descriptive and lack mechanistic proof of relevance. The authors claim that the aim of this study was to elucidate the mechanisms underlying improvement of metabolism by RYGB, but the presented data do unfortunately not provide any mechanistic insights on how metabolism is improved by RYGB. I applaud the authors efforts to describe the differences in gut microbiota between obese patients w/o T2D, but while the data reveal that the surgery reversed a number of disrupted pathways characteristic for T2D, it remains unclear if any of those changes are cause or consequence of the benefits achieved by the surgery. There are hundreds (if not thousands) of studies that report metabolic benefits after RYGB, but what separates studies with higher value is the demonstration that a certain metabolic alteration is indeed causal for the improvement of energy or glucose control after the surgery. Ironically, the authors state that "Therefore, there is a need to move beyond simply profiling the composition of GM communities in order to understand the true nature of host-microbe relationships." While this reviewer fully agrees with this statement, the true nature of host-microbe relationships is unfortunately not addressed in the manuscript.

The authors report that quantities of *Escherichia* (Proteobacteria), *Peptostreptococcaceae* (Firmicutes) and *Barnesiella* (Bacteroidetes) are lower T2D relative to non-diabetic controls, while *Ruminococcus*, *Parabacteroides* and *Bacteroides* had higher relative abundance. Are similar patterns also observed upon RYGB in mice or rats? If so, the effect of these genera on the metabolic effects of the surgery can be directly assessed in mice or rats to assess a causal relationship of these genera with the efficacy of the surgery to improve metabolism.

How do the authors rule out that the observed changes in the host-microbe are mediated by changes in food intake (diet and caloric intake) rather than by the effects of the surgery on weight loss and/or improved glycemic control? Is it possible that the observed changes are mediated by the dietary changes and are unrelated to the surgery? I'm afraid a diet-control group would have been required to answer that question and to exclude any potential bias by the dietary changes after the surgery.

On another note, it would have been interesting to compare changes in the host-microbe between individuals who after a few years regain T2D against those who do not regain T2D. Admittedly, this experimental setup is quite challenging, since we would need groups not only matched for RYGB, but also for body weight.

In summary, I think this is a nice and well written manuscript but the data fall a bit behind my expectation to really get new insights on the potential functional relevance of the identified changes in the microme for the metabolic benefits of the surgery.

Reviewer #2 (Remarks to the Author):

The authors have investigated biological mechanisms underlying type-2 diabetes (T2B) improvement following Roux gastric bypass (RYGB) and vertical sleeve gastrectomy (VSG). They have utilized multi-omics measurements (e.g., microbial species and metabolites) from different sample matrices (e.g., serum, urine, and faecal) and reported significant shifts in omics feature levels as biomarkers, and function level, metabolic pathways. Please see specific

comments:

a) major :

a-1) I would not use the longitudinal term for this study since there are only two time-points, pre- and post- RYGB. If there were three or more time points and the dynamics of changes were investigated, then the longitudinal term would be more appropriate.

a-2) Usually, figure improvements I report as minor; however, for this manuscript, I believed the improvement of the figures could significantly raise the quality of the work. For example, Figures 1 and 2 can be combined and labels font can be adjusted for readability.

a-3) how (statistically) this has been measured “although the percent weight loss was greater after RYGB ($p=0.023$) (Figure 2B)” in lines 175-176.

a-4) Current Figures 2C and 2D do not match with reported significant changes in lines 187-191. Figure 2C might need to be changed to boxplot with showing actual points, otherwise how fruit changes are not significant.

a-5) In statistical test such as PERMENOVA confounding factors' effects should be considered. It is not clear how this has been addressed. I suggest having diet and BMI as confounding factors and do two analyses each by including one of Bariatric Surgery and T2D as a fixed effect. This can be done by a generalized linear model implemented in tools such as Maaslin2 or Tweedieverse R packages. The coefficients of metabolites or other omics features; then can be used for pathway analysis.

a-6) In lines 168-169, Peptostreptococcaceae and Barnesiella are family level. Also, why the associations are reported in family level, species level is more appropriate, as this study has done metagenomics and also species in a genus-level even have different functions and behavior.

a-7) for pathway enrichment analysis using in house method, is not clear how it works. An option would be to use the coefficients described in a-5 and be used via methods for enrichment analysis. Otherwise, clarification on the in-house method would be sufficient.

a-8) Figure 6 is the most important figure I believe. Integration of microbial species and metabolites can be specific to each group and compared in two rows: one for RYGB and one for VSG. Additional analysis would be to include pathways, metabolites, and microbes that are significantly changed post-surgery for each group.

a-9) a summary table of significant changes that have been seen before and what is uniquely found in this study will be very beneficial.

b) minor:

b-1) In the abstract GM need to be spelled out: "GM-host co-metabolism".

b-2) Figure 5, keep the order of subplots in row 1 and row 2 the same for ease of comparison. Also, please make labels (e.g., taxa and KEGG) larger.

b3) Does the order of phylum to species need to be reversed in Figure 3C.

b4) In figure 4C-D, Y-axis labels need to be revised: $-\log_{10}(q\text{-value}) * (\text{sign of } B)$. sign of B is basically $(B/|B|)$.

b5) I would bring a shorter version of Fig 21 and 22 in the main text.

Reviewer #3 (Remarks to the Author):

Summary:

This is a peer review of a Nature Communications manuscript 21-0598-T, entitled "Longitudinal multi-omic phenotyping reveals host-microbe responses to bariatric surgery, glycemic control and obesity" by Penny and colleagues. In general, this is an exploratory basic science manuscript that presents significant novel data regarding urinary, salivary, and gut metabolomic and metagenomic responses in patients undergoing bariatric surgery, specifically Roux-en-Y gastric bypass and Vertical Sleeve Gastrectomy. Additionally, as some individuals were living with diabetes and some not, there are some comparisons/differences that are made between those groups as well before and after surgery. The current study examines patients longitudinally before and 3 months after bariatric surgery. Serum, urine, and fecal metabolic profiles were examined in these individuals. The operations in the study were not randomized, and thus (as expected) more diabetic patients underwent gastric bypass compared to sleeve gastrectomy. In general, the major findings besides presenting this large amount of microbiome and metabolomic data are that there were several gene pathways that appeared to be altered by bariatric surgery with significant overlap to type 2 diabetes. However, there was much less overlap between these diabetes pathways and body mass index. Like other studies, the authors note that the current study suggests that there may be weight-independent mechanisms that are driving resolution of type 2 diabetes in patients following bariatric surgery. Many studies have demonstrated these orally described effects, but the current study suggests these types of weight independent effects by examining microbiome responses. Although this is an interesting and very exciting hypothesis, there is no direct data to support this hypothesis from the manuscript (see comments below). Otherwise, the authors make several notable correlations between various metabolite concentrations and gene pathways. The findings are of significant interest to the microbiome community as well as those scientific investigators working to identify the underlying pathways that are responsible for the dramatic metabolic improvements following bariatric surgery. This is a large amount of data and will surely generate numerous hypotheses that can potentially be tested in additional clinical and pre-clinical models. I applaud the authors for such a comprehensive metabolomic/genomic and microbiome study. Given the large data sets and multiple comparisons, I will have to defer to a biostatistician on the analyses - though on first examination all appear reasonable and adjusted for multiple comparisons and the exploratory nature of the study.

Major Comments:

1. The study overall is an association study that cannot prove any direct mechanisms. This is the major limitation of all, but an inherent limitation of any large data (metabolomic, genomic) study. However, this is one of the largest such studies in bariatric surgical patients that has been done thus far. With the metabolomic and metagenomic data being so broad, however, there are some correlations that the authors note that seem to “make sense” with the changes in microbial gene pathways and corresponding metabolites. Regardless, this study is still limited as flux through these individual pathways cannot be definitely measured without the use of isotopic tracer methodology. This should be noted as a limitation, as many individuals in the metabolomic field do not appreciate this limitation and need for tracer techniques.
2. In this area (microbiome and bariatric surgery) there have been several other smaller studies (in terms of sample size) examining the link between bariatric surgery and the gut microbiome that have been published, though none of them is as large a study as the current manuscript. The real strength of the current manuscript is the preoperative and postoperative measurements in the patients, along with the breadth of metabolomic and metagenomic data that is collected from these individuals.
3. Figure 1 shows correlations between biofluid metabolites and other characteristics. These include dietary scores that were self-reported by the patients. Diet preoperatively and bariatric patients can be extremely variable. As a clinical surgeon, many of these patients will be on a preoperative weight loss regimen, while others may not have any weight loss goal at all. Is there any information on which patients may have been trying to lose weight versus not trying at all. Is there any other information other than the diet scores that is available for these patients? Was the preoperative diet standardized for the patients?
4. How many different surgeons did the operations? Were there any big differences in technique? It is important to briefly describe exactly how the procedure was done – vagal-sparing vs. not, size of pouch, limb lengths for gastric bypass, etc.
5. On lines 239-240 UDCA is discussed...Were patients prescribed ursodiol in the postoperative period? There are important considerations here, especially since the authors note changes in UDCA and its conjugates. A previous study from Vanderbilt University (JCEM 2015, PMID: 2570157) demonstrated that at 1 month individuals post-gastric bypass had increases in UDCA, which is a bacterially-derived secondary bile acid. This is notable because all the patients in that cohort WERE NOT on exogenous UDCA replacement for prophylaxis for gallstones. However, many surgeons use UDCA/actigall/ursodiol for gallstone prophylaxis following bariatric surgery. Can you confirm and specifically note that these patients did or did not receive prophylaxis? If they did receive prophylaxis, then I would remove the data regarding UDCA, as this reflects exogenous administration and not necessarily endogenous synthesis from the gut microbiome.
6. Were some patients required to lose weight and others not prior to surgery? This is a surgical consideration that could affect weight loss trajectory and especially diet preoperatively. Do you know what the preoperative weight loss was from the time of initial consultation – this would give an indication of how much body weight may have changed secondary to other dietary effects that were not necessarily measured in the current data set.

Minor Comments:

1. How was the determination made for impaired glucose tolerance? How is that defined clinically? It would be good to draw this distinction between normal and frank diabetes.
2. Is there any information on the pharmacotherapies that patients may have been receiving preoperatively that could affect gut microbiome or other metabolic pathways?

3. Results – it's somewhat unclear on lines 86 to 88 how many normal glucose tolerance individuals there are. It states that 42 were diabetic and 11 had impaired glucose tolerance, can you include the remainder of those individuals which should add up to 81 for clarity.

Reviewer #4 (Remarks to the Author):

This paper describes a multiomics study targeting patients with diabetes (T2D) and obesity (as determined by the BMI). The novelty is that they studied the effect of bariatric surgery on the metabolome and in the gut microbiota. The cohort size is adequate considering the complexity of the inclusion criteria and the experiments have been carefully planned and are technically sound. The manuscript reads very well and the discussion and the conclusions are fully supported by the experimental evidences found. Overall, this is an excellent contribution and I only have some minor comments for the author's consideration:

1) The effect of the medication in the T2D patients is not properly addressed. In one sentence it is stated that Metformin was found in urine. This information was collected (shown in the supplementary material) but not discussed in the main text.

2) Also, one of the main findings is the alteration of the cholic acids after bariatric surgery. Yet, gallstone formation is a common complication of this procedures. Moreover, cholecystectomy was NOT an exclusion criteria.

3) Regarding the changes in the GM after the surgical interventions, was the diet control by the doctor or a nutritionist? How about probiotic substances, were they forbidden from the diet?

4) Figure 2A is cited before Figure 1.

Reviewer Responses

Reviewers' comments:

Reviewer #1 (Remarks to the Author):

In this manuscript, Penney and colleagues assessed, using a multi-omics approach, the similarities and differences in host-microbe responses to bariatric surgery, glycaemic control and obesity. The authors report that bariatric surgery reversed a number of changes in the microbe that are characteristic for T2D.

The manuscript is well written and easy to understand. It deals with an interesting and important topic, the role of the host-microbe in the metabolic effects of bariatric surgery and its effects on improving energy and glucose metabolism.

A key weakness of the here presented work is that authors fail to provide any mechanistic evidence that links the reported host-microbe responses to the metabolic effects of bariatric surgery, glycaemic control and obesity. The results, despite interesting, are quite descriptive and lack mechanistic proof of relevance. The authors claim that the aim of this study was to elucidate the mechanisms underlying improvement of metabolism by RYGB, but the presented data do unfortunately not provide any mechanistic insights on how metabolism is improved by RYGB. I applaud the authors efforts to describe the differences in gut microbiota between obese patients w/o T2D, but while the data reveal that the surgery reversed a number of disrupted pathways characteristic for T2D, it remains unclear if any of those changes are cause or consequence of the benefits achieved by the surgery. There are hundreds (if not thousands) of studies that report metabolic benefits after RYGB, but what separates studies with higher value is the demonstration that a certain metabolic alteration is indeed causal for the improvement of energy or glucose control after the surgery. Ironically, the authors state that “Therefore, there is a need to move beyond simply profiling the composition of GM communities in order to understand the true nature of host-microbe relationships.” While this reviewer fully agrees with this statement, the true nature of host-microbe relationships is unfortunately not addressed in the manuscript.

Response:

The aim of this study was to understand associations between omics data that may help explain or suggest mechanisms involved in the pathology of obesity and diabetes and those that bariatric surgery alters to reverse these. The sentence highlighted by the reviewer regarding moving “beyond simply profiling the composition of GM communities” is intended to highlight the fact that this study incorporates multiple inter-related profiles. i.e. we combine gut microbiome, metabolite, dietary and clinical data sets. A number of prior studies have assessed one or other of these datasets in bariatric surgery patients or diabetics, but few have combined all, which we believe has allowed greater insights into the influence of each and how they relate to each other. Furthermore, we have incorporated data comparing

the influence of glycaemic control /diabetes as well as obesity. A number of studies that have assessed bariatric surgery previously have not also assessed the influence on diabetes.

Whilst we fully acknowledge that performing mechanistic studies in animal models would strengthen this work this would entail a large body of work and we believe that this is beyond the scope of this single study. We plan to perform this as future work in a humanised mouse model once funding is secured.

The authors report that quantities of Escherichia (Proteobacteria), Peptostreptococcaceae (Firmicutes) and Barnesiella (Bacteroidetes) are lower T2D relative to non-diabetic controls, while Ruminococcus, Parabacteroides and Bacteroides had higher relative abundance. Are similar patterns also observed upon RYGB in mice or rats? If so, the effect of these genera on the metabolic effects of the surgery can be directly assessed in mice or rats to assess a causal relationship of these genera with the efficacy of the surgery to improve metabolism.

Response:

There are indeed several correlates between the current study and previous studies in rodent models in terms of the microbiome and others have studied microbiota changes in mice or rats post bariatric surgery. While there are some similarities, as with all animal models there are also significant differences in the composition of gut communities and their response to surgery compared to humans. A significant strength of this study is that it has been performed in humans. As stated above, whilst we fully acknowledge that performing mechanistic studies in animal models would strengthen this work we believe that this is beyond the scope of this single study and plan to perform this as future work once funding is secured.

We have added the following sentence to the discussion section to address the referee's point and highlight the consistency of the impact of T2D on the microbiome:

“Consistent with our observations in the current study, Ruminococcus and Parabacteroides have a higher relative abundance in T2D rodent models compared to non-diabetic animals^{1,2} and Ruminococcus has been shown to exert pro-inflammatory effects and also to promote pathogenesis in type 1 diabetes³. Similarly, lower relative concentrations of Barnesiella have been associated with T2D in rodent models⁴. In contrast, the role of Bacteroides is less clear with some studies reporting lower abundance of Bacteroides in T2D¹.”

- 1. Yang R et al. Genistein ameliorates inflammation and insulin resistance through mediation of gut microbiota composition in type 2 diabetic mice. Eur J Nutr. Jun 2021;60(4):2155-2168.*
- 2. Zheng S et al. Oleuropein Ameliorates Advanced Stage of Type 2 Diabetes in db/db Mice by Regulating Gut Microbiota. Nutrients. Jun 22 2021;13(7)*
- 3. Krych L et al. Gut microbial markers are associated with diabetes onset, regulatory imbalance, and IFN-gamma level in NOD mice. Gut microbes. 2015;6(2):101-9.*
- 4. Nam Y et al. Heat-Killed Lactiplantibacillus plantarum LRCC5314 Mitigates the Effects of Stress-Related Type 2 Diabetes in Mice via Gut Microbiome Modulation. Journal of microbiology and biotechnology. Mar 28 2022;32(3):324-332.*

How do the authors rule out that the observed changes in the host-microbe are mediated by changes in food intake (diet and caloric intake) rather than by the effects of the surgery on weight loss and/or improved glycemic control? Is it possible that the observed changes are mediated by the dietary changes and are unrelated to the surgery? I'm afraid a diet-control group would have been required to answer that question and to exclude any potential bias by the dietary changes after the surgery.

Response:

We agree that a number of the host-microbe changes that occur after bariatric surgery are mediated via changes in the diet. Indeed, dietary changes are an integral mechanism, amongst others, that mediate the effects of bariatric surgery, including weight loss and also influencing the gut microbes. In keeping with this, we have tried to highlight in the manuscript the large changes to the diet that occur after bariatric surgery. For example, these are shown in Figure 2. We also discuss these changes at several points in the results and discussion. This includes exploring bacteria and metabolites that correlate strongly with dietary factors. We have now also moved Figure 21 from the supplementary data into the main manuscript which helps to highlight these correlations (now Figure 7)

While a diet-control group would have been an excellent addition to the study, one of the interesting aspects of bariatric surgery is the ability of patients to sustain such a large reduction in the caloric intake without the sensation of extreme hunger due to a number of mechanisms such as alterations in gut hormones. Sustaining this change in diet without bariatric surgery is extremely difficult, indeed all participants in this study had failed attempts at prior dieting. Patients in this study lost approximately 22-25kg at 3 months post operation. Whereas people can be expected to lose approximately 5kg after a 6-month supervised multicomponent Tier3 weight management programme¹. Thus, highlighting that fact that having a diet-control group with similar changes to both their diet and weight to that of a bariatric surgery cohort over a 3-month period would be extremely challenging, if not impossible.

One other aspect to consider is the fact that in a mouse model of caloric restriction versus bariatric surgery, both weight loss dependent and independent metabolic changes were observed after Roux-en-Y bariatric surgery².

- 1. Alkharaji M et al. Tier 3 specialist weight management service and pre-bariatric multicomponent weight management programmes for adults with obesity living in the UK: A systematic review. *Endocrinol Diabetes Metab.* Jan 2019;2(1):e00042.*
- 2. Seyfried F et al. Urinary Phenotyping Indicates Weight Loss-Independent Metabolic Effects of Roux-en-Y Gastric Bypass in Mice. *Journal of Proteome Research*, (2013). 12(3), 1245–1253.*

On another note, it would have been interesting to compare changes in the host-microbe between individuals who after a few years regain T2D against those who do not regain T2D. Admittedly, this experimental setup is quite challenging, since we would need groups not only matched for RYGB, but also for body weight.

Response:

Again, we agree with the reviewer that this would be very interesting. However, as the reviewer highlights this would be a challenging experimental setup and is beyond the scope of this single study.

In summary, I think this is a nice and well written manuscript but the data fall a bit behind my expectation to really get new insights on the potential functional relevance of the identified changes in the microbiome for the metabolic benefits of the surgery.

Response:

We hope we have addressed the comments of the reviewer and point again to the value of the study, which lies in harnessing this multi-omic approach, together with dietary data and clinical data to generate comprehensive profiles of the effect of Bariatric surgery on the intersecting impacts of obesity and diabetes. Whilst detailed elucidation of specific mechanistic pathways was beyond the scope of the current study, we point to several new avenues that warrant further investigation.

Reviewer #2 (Remarks to the Author):

The authors have investigated biological mechanisms underlying type-2 diabetes (T2B) improvement following Roux gastric bypass (RYGB) and vertical sleeve gastrectomy (VSG). They have utilized multi-omics measurements (e.g., microbial species and metabolites) from different sample matrices (e.g., serum, urine, and faecal) and reported significant shifts in omics feature levels as biomarkers, and function level, metabolic pathways. Please see specific

comments:

a) major :

a-1) I would not use the longitudinal term for this study since there are only two time-points, pre- and post- RYGB. If there were three or more time points and the dynamics of changes were investigated, then the longitudinal term would be more appropriate.

Response:

We thank the reviewer for this comment and have modified the language used in the manuscript appropriately.

a-2) Usually, figure improvements I report as minor; however, for this manuscript, I believed the improvement of the figures could significantly raise the quality of the work. For example, Figures 1 and 2 can be combined and labels font can be adjusted for readability.

Response:

We have worked to improve the figures, including combining Figures 1 & 2 as suggested. A number of the figures contain large amounts of data, but we highlight that the figures are available in high resolution to allow readability on zooming in.

a-3) how (statistically) this has been measured “although the percent weight loss was greater after RYGB (p=0.023) (Figure 2B)” in lines 175-176.

Response:

This was measured using the Mann-Whitney U test. We have clarified this in the methods section where we describe univariate analysis by adding that this was also applied to clinical data.

“Due to the non-parametric nature of the results, differences between paired samples pre/post intervention in clinical data, quantified metabolites and within the gut microbiota were assessed for significance using the Wilcoxon Rank test (two-sided). Differences in non-paired data were assessed using the Mann-Whitney U test (two-sided).”

a-4) Current Figures 2C and 2D do not match with reported significant changes in lines 187-191. Figure 2C might need to be changed to boxplot with showing actual points, otherwise how fruit changes are not significant.

Response:

We thank the reviewer for highlighting this point. As this dietary data was paired data this was assessed using the Wilcoxon Signed Rank test (as described above). Although on the previous figure there appeared to be large changes to the median value of a number of dietary substrates such as fruit, it did not show outliers or the distribution of the data. For example, there were some participants that had no fruit pre-operatively. As suggested by the reviewer, we have now changed this figure to a boxplot design which shows the distribution of the data in much greater detail.

a-5) In statistical test such as PERMANOVA confounding factors' effects should be considered. It is not clear how this has been addressed. I suggest having diet and BMI as confounding factors and do two analyses each by including one of Bariatric Surgery and T2D as a fixed effect. This can be done by a generalized linear model implemented in tools such as Maaslin2 or Tweedieverse R packages. The coefficients of metabolites or other omics features; then can be used for pathway analysis.

Response:

We thank the reviewer for highlighting the influence of confounding factors.

At the initial stage in analysis of the results PERMANOVA / OPLS-DA etc were used to catalogue the gross overall effects of surgery or T2D vs non-T2D. We accept that these effects are influenced by dietary and BMI changes respectively. But we suggest that these changes are part of the overall effect of RYGB or VSG surgery and are causally related, not confounders, and as such did not want to eliminate their effects from this stage of the analysis. For example, dietary changes after bariatric surgery are causally related to weight loss and hence cannot be classified as a true confounder. Similarly, with respect to diabetes, both weight and non-weight related mechanisms are known to be causally related to type-2 diabetes. Hence, again, weight is not a true confounder as it causally related to the outcome. Furthermore, we also highlight the fact that BMI was not significantly different between T2D / non-T2D participants at baseline.

Nevertheless, we fully agree that assessing which components of these interventions influence the outcomes, including weight and glycaemic control, is of vital importance. Indeed, this is a key part of the subsequent analysis and discussion (we refer to these factors as covariates as opposed to confounders).

After cataloguing the changes between groups in single datasets at the initial stage in analysis we then performed extensive further analyses to explore how these changes related to different data sets and covariates / mechanisms using DIABLO and Spearman's

correlations in an effort to assess which covariates / mechanisms influenced which aspects of the microbiota / metabolome.

We have now also added additional results to the manuscript having further isolated the effects of different covariates / mechanisms after adjusting for them, as suggested by the reviewer. We have performed this by adding partial correlation analysis to the manuscript results. In the analysis of correlations to HbA1c we have adjusted for weight and BMI to help isolate weight-independent markers of glycaemic control using partial correlations. Similarly, we have performed partial correlations to BMI adjusting for HbA1c to help reduce the influence of changes to diabetic control after weight loss. We thank the reviewer for highlighting this point and hope that this addition has strengthened the manuscript.

a-6) In lines 168-169, Peptostreptococcaceae and Barnesiella are family level. Also, why the associations are reported in family level, species level is more appropriate, as this study has done metagenomics and also species in a genus-level even have different functions and behaviour.

Response:

The analyses were performed at all taxonomic levels including species and genus level. In the manuscript changes predominantly described at the genus level. This is firstly to improve the clarity of the results. If all the species were outlined this would be extremely long. In addition, while metagenomics is purported to allow species-level analysis it is becoming clear from recent studies that this is not the case when using tools such as Kraken, MetaPhlan and Centrifuge. That is, these tools often do not differentiate closely related species and count them more than once, leading to inflated species-level abundance data (e.g. PMID:32436839). Consequently, we have described the data at genus level, which allowed us to demonstrate differences between our groups of interest.

Regarding the specific points of Peptostreptococcaceae and Barnesiella. Barnesiella is at the genus taxonomic level. Peptostreptococcaceae in this context represented a novel unclassified genus. This has been clarified in the text as Peptostreptococcaceae unclassified.

a-7) for pathway enrichment analysis using in house method, is not clear how it works. An option would be to use the coefficients described in a-5 and be used via methods for enrichment analysis. Otherwise, clarification on the in-house method would be sufficient.

Response:

Functional annotation to KEGG pathways was carried out by mapping centroid sequences to the eggNOG-Mapper database. Integration of datasets (e.g. metabolite, microbiota, KEGG etc) was performed using the Data Integration Analysis for Biomarker discovery using Latent cOmponents (DIABLO)¹, implemented through the mixOmics² package in R.

Analysis of correlations to BMI and HbA1c was performed using Spearman's correlations with multiple testing corrections and as described above, we have now also added partial Spearman's correlations to the analysis to help adjust for covariates.

1. Singh A et al. DIABLO: an integrative approach for identifying key molecular drivers from multi-omics assays. *Bioinformatics*. Sep 1 2019;35(17):3055-3062.
2. Rohart F et al. mixOmics: An R package for 'omics feature selection and multiple data integration. *PLoS Comput Biol*. Nov 2017;13(11):e1005752.

a-8) Figure 6 is the most important figure I believe. Integration of microbial species and metabolites can be specific to each group and compared in two rows: one for RYGB and one for VSG. Additional analysis would be to include pathways, metabolites, and microbes that are significantly changed post-surgery for each group.

Response:

We thank the reviewer for this suggestion. We have attempted this analysis, however, splitting the data into these groups significantly reduces the n number and hence the power of the tests. We found far fewer correlations: of those seen many were similar, but we were concerned that any correlations not seen were due to a type 2 statistical error rather than any true differences between the subgroups. In addition, as described above we have added two further panels to Figure 6 to better control for covariates / competing mechanisms using partial correlations. While differences between the bariatric surgery procedures have already been explored using DIABLO in Figure 5 and supplemental R4. There is a danger that adding further analyses overcomplicates the analysis, taking away from the key messages.

a-9) a summary table of significant changes that have been seen before and what is uniquely found in this study will be very beneficial.

Response:

Given the large breath of metabolic, metagenomic, dietary and clinical data this would be challenging to condense into a summary table form.

The overwhelming disruption to the GM and human metabolome caused by bariatric surgery and its functional importance are only just being defined. Few studies exploring host-microbe interactions have occurred in human cohorts, with most studies focussing on either the microbiome or the metabolome. No studies have integrated multi-biofluid analysis to give a comprehensive metabolic insight and importantly none have analysed urine, which gives important insights into gut-host co-metabolism following bariatric surgery. In this study we have integrated extensive multi-biofluid metabolic analyses across multiple biological matrices with shotgun metagenomics (taxonomic and functional profiling) of the GM in both i) T2D versus non-diabetic BMI matched participants at baseline and ii) participants longitudinally profiled pre and post RYGB and vertical sleeve gastrectomy (VSG). This integrative analysis has provided a comprehensive atlas of host-microbe responses to bariatric surgery, glycaemic control and BMI at the systems level and unique insights into the influence of each in humans. Profiling patients with and without T2D has enabled us to dissect the effects of weight-loss over improved glycaemic control. Additionally, we found significant differences between the contrasting bariatric procedures. We identified a number of bacterial taxa and metabolic pathways that are altered following RYGB, but not VSG. These mainly related to increased amino acid metabolism pathways within the GM, with corresponding increases in urinary metabolites associated with microbial protein degradation. Whereas changes in BA metabolism did not appear to be primarily influenced by altered GM.

As stated by reviewer 3 we hope that: “The findings are of significant interest to the microbiome community as well as those scientific investigators working to identify the underlying pathways that are responsible for the dramatic metabolic improvements following

bariatric surgery. This is a large amount of data and will surely generate numerous hypotheses that can potentially be tested in additional clinical and pre-clinical models.”

b) minor:

b-1) In the abstract GM need to be spelled out: “GM-host co-metabolism”.

Response:

This has been amended.

b-2) Figure 5, keep the order of subplots in row 1 and row 2 the same for ease of comparison. Also, please make labels (e.g., taxa and KEGG) larger.

Response:

Having row 2 read from right to left allows the main plots (C+F) to be larger and improve readability. However, we would be happy to be guided by the editorial team if they agree that row 2 should read from left to right.

b3) Does the order of phylum to species need to be reversed in Figure 3C.

Response:

The labels represent the nodes in the figure and are correctly ordered.

b4) In figure 4C-D, Y-axis labels need to be revised: $-\log_{10}(\text{q-value}) * (\text{sign of B})$. sign of B is basically $(B/|B|)$.

Response:

The labels have been revised accordingly.

b5) I would bring a shorter version of Fig 21 and 22 in the main text.

Response:

On the suggestion of the reviewer, we have brought Figure 21 into the main text.

Reviewer #3 (Remarks to the Author):

Reviewer 3

Summary:

This is a peer review of a Communications Medicine manuscript 21-0598-T, entitled "Longitudinal multi-omic phenotyping reveals host-microbe responses to bariatric surgery, glycemic control and obesity" by Penny and colleagues. In general, this is an exploratory basic science manuscript that presents significant novel data regarding urinary, salivary, and gut metabolomic and metagenomic responses in patients undergoing bariatric surgery, specifically Roux-en-Y gastric bypass and Vertical Sleeve Gastrectomy.

Additionally, as some individuals were living with diabetes and some not, there are some comparisons/differences that are made between those groups as well before and after surgery. The current study examines patients longitudinally before and 3 months after bariatric surgery. Serum, urine, and fecal metabolic profiles were examined in these individuals. The operations in the study were not randomized, and thus (as expected) more diabetic patients underwent gastric bypass compared to sleeve gastrectomy.

In general, the major findings besides presenting this large amount of microbiome and metabolomic data are that there were several gene pathways that appeared to be altered by bariatric surgery with significant overlap to type 2 diabetes. However, there was much less overlap between these diabetes pathways and body mass index. Like other studies, the authors note that the current study suggests that there may be weight-independent mechanisms that are driving resolution of type 2 diabetes in patients following bariatric surgery. Many studies have demonstrated these orally described effects, but the current study suggests these types of weight independent effects by examining microbiome responses. Although this is an interesting and very exciting hypothesis, there is no direct data to support this hypothesis from the manuscript (see comments below). Otherwise, the authors make several notable correlations between various metabolite concentrations and gene pathways.

The findings are of significant interest to the microbiome community as well as those scientific investigators working to identify the underlying pathways that are responsible for the dramatic metabolic improvements following bariatric surgery. This is a large amount of data and will surely generate numerous hypotheses that can potentially be tested in additional clinical and pre-clinical models. I applaud the authors for such a comprehensive metabolomic/genomic and microbiome study. Given the large data sets and multiple comparisons, I will have to defer to a biostatistician on the analyses - though on first examination all appear reasonable and adjusted for multiple comparisons and the exploratory nature of the study.

Major Comments:

1. The study overall is an association study that cannot prove any direct mechanisms. This is the major limitation of all, but an inherent limitation of any large data (metabolomic, genomic) study. However, this is one of the largest such studies in bariatric surgical patients that has been done thus far. With the metabolomic and metagenomic data being so broad, however, there are some correlations that the authors note that seem to “make sense” with the changes in microbial gene pathways and corresponding metabolites. Regardless, this study is still limited as flux through these individual pathways cannot be definitely measured without the use of isotopic tracer methodology. This should be noted as a limitation, as many individuals in the metabolomic field do not appreciate this limitation and need for tracer techniques.

Response:

We thank the reviewer for the comment regarding tracer methodology. We have added a comment to the manuscript conclusion in relation to this.

“Although this is one of the largest studies to date integrating metagenomic and metabolomic data from bariatric surgery patients and presents several novel avenues for future study, we

acknowledge our approach is not without limitations. Specifically, metagenomics provides functional predictions with respect to microbial metabolite production. To allow us to fully elucidate the contribution of the microbiota to the metabolites we detect, a standardized dietary intervention incorporating one or more stable-isotope-labelled substrates would be required in our cohort. This approach would not only allow us to pinpoint those metabolites produced via microbial biotransformations but would also allow us to incorporate metabolic flux analysis into our metagenomic analyses.”

2. In this area (microbiome and bariatric surgery) there have been several other smaller studies (in terms of sample size) examining the link between bariatric surgery and the gut microbiome that have been published, though none of them is as large a study as the current manuscript. The real strength of the current manuscript is the preoperative and postoperative measurements in the patients, along with the breath of metabolomic and metagenomic data that is collected from these individuals.

Response:

We thank the reviewer for this comment and are in agreement and have highlighted this in the discussion.

3. Figure 1 shows correlations between biofluid metabolites and other characteristics. These include dietary scores that were self-reported by the patients. Diet preoperatively and bariatric patients can be extremely variable. As a clinical surgeon, many of these patients will be on a preoperative weight loss regimen, while others may not have any weight loss goal at all. Is there any information on which patients may have been trying to lose weight versus not trying at all. Is there any other information other than the diet scores that is available for these patients? Was the preoperative diet standardized for the patients?

Response:

Prior to recruitment all patients had attended and complied with a Tier 3 intense supervised multidisciplinary diet and lifestyle program for a minimum of 6 months. Patients were no longer undergoing any specific intensive weight loss programme at the time of recruitment. After recruitment participants were placed on a ‘liver shrinking’ diet for 2 weeks prior to surgery. This consisted of 100g carbohydrate per day, low fat intake and limiting overall intake to 800-1000 kcal / day. Participants were given an extensive meal plan to help achieve this.

We have added these clinical details to the Supplementary Information.

4. How many different surgeons did the operations? Were there any big differences in technique? It is important to briefly describe exactly how the procedure was done – vagal-sparing vs. not, size of pouch, limb lengths for gastric bypass, etc.

Response:

We thank the reviewer for highlighting this important point and agree that the techniques used for surgical interventions should be clearly reported in clinical trials. We have added the details below to the Supplementary Information:

The procedures were performed by 5 consultant surgeons from within the same department. All procedures were standardised and performed in the same manner as described in brief below.

RYGB:

Gastric Pouch - 15 to 30cc gastric pouch created using laparoscopic staplers. Sized using a 34Fr Ewald tube.

Biliopancreatic Limb: 75cm.

Roux Limb: 100cm.

Gastrojejunostomy - Side to side anastomosis using 30mm stapler, enterotomy closed with continuous 3-0 vicryl followed by a second layer of interrupted 3-0 vicryl.

Underwater leak test used to check anastomosis.

Jejunal-Jejunal anastomosis - Side to side using 45mm stapler, enterotomy closed with continuous 3-0 vicryl.

Mesenteric and Petersen's defects were closed with IFA bond.

VSG:

Gastric Sleeve formed over a 34Fr orogastric tube.

Stapling commenced 3cm proximal from pylorus.

4-5 firings of 45/60mm staplers with Seamguard staple line re-enforcement.

Underwater leak test used to check staple line.

5. On lines 239-240 UDCA is discussed... Were patients prescribed ursodiol in the postoperative period? There are important considerations here, especially since the authors note changes in UDCA and its conjugates. A previous study from Vanderbilt University (JCEM 2015, PMID: PMC4570157) demonstrated that at 1 month individuals post-gastric bypass had increases in UDCA, which is a bacterially-derived secondary bile acid. This is notable because all the patients in that cohort WERE NOT on exogenous UDCA replacement for prophylaxis for gallstones. However, many surgeons use UDCA/actigall/ursodiol for gallstone prophylaxis following bariatric surgery. Can you confirm and specifically note that these patients did or did not receive prophylaxis? If they did receive prophylaxis, then I would remove the data regarding UDCA, as this reflects exogenous administration and not necessarily endogenous synthesis from the gut microbiome.

Response:

We thank the reviewer for highlighting this issue. Yes, patients who had not previously undergone a cholecystectomy were prescribed UDCA for gallstone prophylaxis. We have added a comment to the text at this point highlighting that the changes to UDCA, although seen by others in individuals not given UDCA as highlighted by the reviewer, are likely to be predominantly due to exogenous administration of UDCA.

6. Were some patients required to lose weight and others not prior to surgery? This is a surgical consideration that could affect weight loss trajectory and especially diet preoperatively. Do you know what the preoperative weight loss was from the time of initial consultation – this would give an indication of how much body weight may have changed secondary to other dietary effects that were not necessarily measured in the current data set.

Response:

As detailed above, prior to recruitment all patients had attended and complied with a Tier 3 intense supervised multidisciplinary diet and lifestyle program for a minimum of 6 months. Patients were no longer undergoing any specific intensive weight loss programme at the time of recruitment. After recruitment participants were placed on a 'liver shrinking' diet for 2 weeks prior to surgery. This consisted of 100g carbohydrate per day, low fat intake and

limiting overall intake to 800-1000 kcal / day. Participants were given an extensive meal plan to help achieve this. Unfortunately, the data measuring weight changes from time of recruitment to the date of surgery is incomplete and precludes formal analysis.

These details have been added to the Supplementary Information.

Minor Comments:

1. How was the determination made for impaired glucose tolerance? How is that defined clinically? It would be good to draw this distinction between normal and frank diabetes.

Response:

In line with international guidelines, impaired glucose tolerance (IGT) was defined as participants with an HbA1c 42-47 mmol/mol. T2D was defined as HbA1c \geq 48mmol/mol. This has now been clarified in the recruitment section of the methods. T2D analysis compared patients with 'frank' T2D to non-diabetics and did not include IGT patients. Correlation analysis to HbA1c included all patients, including IGT, to allow inclusion of a full range of glycaemic control into the analysis.

2. Is there any information on the pharmacotherapies that patients may have been receiving preoperatively that could affect gut microbiome or other metabolic pathways?

Response:

Information on diabetic medication use is available in the Supplementary Information. Statin use was near universal across all study participants. We discuss the impact of metformin on the gut microbiome in the discussion:

“Some of the differences in this cohort, such as lower Clostridium bartlettii (aka Intestinibacter bartlettii) levels relative to non-diabetic controls, are in keeping with changes due to metformin use (86% metformin use in T2D group, see Supplemental Information R1). However, other differences in T2D participants such as lower Escherichia coli were apparent despite metformin’s well characterised action in increasing its relative abundance.”

We have also added the following:

“Metformin is known to exert an influence on the serum metabolite profile and is associated with lower concentrations of lipoproteins, N-acetylated glycoproteins, lactate and glucose, along with increased relative concentrations of TMAO and 3-hydroxybutyrate¹. Thus, metformin shifts the serum profile towards a non-diabetic state and most of the metabolic differences observed in the current study do not appear to be differentially modulated by Metformin.” In future experiments it would be good to define the exact impact of metformin on the metabolic parameters measured in this study.

1. *Huo T, et al. Metabonomic study of biochemical changes in the serum of type 2 diabetes mellitus patients after the treatment of metformin hydrochloride. J Pharm Biomed Anal. 2009 May 1;49(4):976-82.*

3. Results – it's somewhat unclear on lines 86 to 88 how many normal glucose tolerance individuals there are. It states that 42 were diabetic and 11 had impaired glucose tolerance, can you include the remainder of those individuals which should add up to 81 for clarity.

Response:

We thank the reviewer for highlighting this. This has been amended.

Reviewer #4 (Remarks to the Author):

This paper describes a multiomics study targeting patients with diabetes (T2D) and obesity (as determined by the BMI). The novelty is that they studied the effect of bariatric surgery on the metabolome and in the gut microbiota. The cohort size is adequate considering the complexity of the inclusion criteria and the experiments have been carefully planned and are technically sound. The manuscript reads very well and the discussion and the conclusions are fully supported by the experimental evidences found. Overall, this is an excellent contribution and I only have some minor comments for the author's consideration:

Minor Comments:

1) The effect of the medication in the T2D patients is not properly addressed. In one sentence it is stated that Metformin was found in urine. This information was collected (shown in the supplementary material) but not discussed in the main text.

Response:

*We thank the reviewer for highlighting the importance of medication. We direct the reviewer to the paragraph where we discuss the impact of metformin on the gut microbiota: "Some of the differences in this cohort, such as lower *Clostridium bartlettii* (aka *Intestinibacter bartlettii*) levels relative to non-diabetic controls, are in keeping with changes due to metformin use (86% metformin use in T2D group, see Supplemental Information R1). However, other differences in T2D participants such as lower *Escherichia coli* were apparent despite metformin's well characterised action in increasing its relative abundance."*

As mentioned above, we have now added to the manuscript that metformin also influences the serum metabolome and serves to reduce LDL and VLDL lipoproteins, glucose and acetylated glycoproteins. Thus, metformin shifts the serum profile towards a non-diabetic state. Since the groups for individuals with and without type 2 diabetes were compared directly and there are still metabolic differences relating to diabetes, we can be confident that metformin is not substantially obscuring the relationship between the serum metabolome and diabetes.

Finally, we have also added to the results that:

"The improvement in lipid profile was seen despite near universal use of statins preoperatively."

2) Also, one of the main findings is the alteration of the cholic acids after bariatric surgery. Yet, gallstone formation is a common complication of this procedures. Moreover, cholecystectomy was NOT an exclusion criteria.

Response:

We are not aware that cholecystectomy has any influence on the composition of the bile pool (although this would be an interesting study). However, we do acknowledge that a number of

these patients were prescribed UDCA exogenously post operatively to prevent gallstone formation and this is the likely predominant cause of the changes seen in UDCA and it's conjugates, which we have now highlighted in the manuscript.

3) Regarding the changes in the GM after the surgical interventions, was the diet control by the doctor or a nutritionist? How about probiotic substances, were they forbidden from the diet?

Response:

Patients were given advice on the texture of foods to consume postoperatively. This consisted of a liquid diet for 10 days, puree diet for 3-4 weeks, followed by a soft diet for approximately 6 weeks before resuming a normal texture diet. Patients were given advice by the dietetics team to follow a healthy balanced diet with sufficient quantities of protein and to consume multivitamin & mineral supplements lifelong. Probiotics were not forbidden, but we are not aware of any patients consuming them.

We have added the above information on dietary advice to the Supplemental Information.

4) Figure 2A is cited before Figure 1.

Response:

Figures 1 & 2 have now been changed and are appropriately cited in the text.

REVIEWERS' COMMENTS:

Reviewer #1 (Remarks to the Author):

I carefully went over the authors comments. Unfortunately, the authors didn't address any of my concerns. While I consider mechanistic studies essential to understand and interpret the here presented data, the authors consider these as beyond the scope of the here presented work. The data remain very descriptive. The studies unfortunately still lack functional proof of causality.

Reviewer #2 (Remarks to the Author):

The authors have investigated biological mechanisms underlying type-2 diabetes (T2B) improvement following Roux gastric bypass (RYGB) and vertical sleeve gastrectomy (VSG). They have utilized multi-omics measurements (e.g., microbial species and metabolites) from different sample matrices (e.g., serum, urine, and faecal) and reported significant shifts in omics feature levels as biomarkers, and function level, metabolic pathways. Please see specific

comments:

a) major :

a-1) I would not use the longitudinal term for this study since there are only two time-points, pre- and post- RYGB. If there were three or more time points and the dynamics of changes were investigated, then the longitudinal term would be more appropriate.

Response:

We thank the reviewer for this comment and have modified the language used in the manuscript appropriately.

a-2) Usually, figure improvements I report as minor; however, for this manuscript, I believed the improvement of the figures could significantly raise the quality of the work. For example, Figures 1 and 2 can be combined, and labels font can be adjusted for readability.

Response:

We have worked to improve the figures, including combining Figures 1 & 2 as suggested. A number of the figures contain large amounts of data, but we highlight that the figures are available in high resolution to allow readability on zooming in.

Reviewer: I found it to be very difficult to check the figures if they are not in the main text with their number and caption. I feel figures still need more work to get publishable equality. For example in figure 1, do we need all this space for that amount of information presented by subplots?

a-3) how (statistically) this has been measured “although the percent weight loss was greater after RYGB ($p=0.023$) (Figure 2B)” in lines 175-176.

Response:

This was measured using the Mann-Whitney U test. We have clarified this in the methods section where we describe univariate analysis by adding that this was also applied to clinical data.

“Due to the non-parametric nature of the results, differences between paired samples pre/post intervention in clinical data, quantified metabolites and within the gut microbiota were assessed for significance using the Wilcoxon Rank test (two-sided). Differences in non-paired data were assessed using the Mann-Whitney U test (two-sided).”

Reviewer: No further comments, thanks for the clarifications.

a-4) Current Figures 2C and 2D do not match with reported significant changes in lines 187-191. Figure 2C might need to be changed to a boxplot showing actual points, otherwise how fruit changes are not significant.

Response:

We thank the reviewer for highlighting this point. As this dietary data was paired data this was assessed using the Wilcoxon Signed Rank test (as described above). Although on the previous figure there appeared to be large changes to the median value of a number of dietary substrates such as fruit, it did not show outliers or the distribution of the data. For example, there were some participants that had no fruit pre-operatively. As suggested by the reviewer, we have now changed this figure to a boxplot design which shows the distribution of the data in much greater detail.

Reviewer: Thanks for the improvement, if Figure 2C what sugar is not significant? Points over box plots help to see what caused the fact that box plots for sugar pre- and post-surgery are not significantly different even boxplot shows that.

a-5) In statistical tests such as PERMANOVA confounding factors' effects should be considered. It is not clear how this has been addressed. I suggest having diet and BMI as confounding factors and doing two analyses each by including one of Bariatric Surgery and T2D as a fixed effect. This can be done by a generalized linear model implemented in tools such as Maaslin2 or Tweedieverse R packages. The coefficients of metabolites or other omics features; then can be used for pathway analysis.

Response:

We thank the reviewer for highlighting the influence of confounding factors.

At the initial stage in analysis of the results PERMANOVA / OPLS-DA etc were used to catalogue the gross overall effects of surgery or T2D vs non-T2D. We accept that these effects are influenced by dietary and BMI changes respectively. But we suggest that these changes are part of the overall effect of RYGB or VSG surgery and are causally related, not confounders, and as such did not want to eliminate their effects from this stage of the analysis. For example, dietary changes after bariatric surgery are causally related to weight loss and hence cannot be classified as a true confounder. Similarly, with respect to diabetes, both weight and non-weight related mechanisms are known to be causally related to type-2 diabetes. Hence, again, weight is not a true confounder as it causally related to the outcome. Furthermore, we also highlight the fact that BMI was not significantly different between T2D / non-T2D participants at baseline.

Nevertheless, we fully agree that assessing which components of these interventions influence the outcomes, including weight and glycaemic control, is of vital importance. Indeed, this is a key part of the subsequent analysis and discussion (we refer to these factors as covariates as opposed to confounders).

After cataloguing the changes between groups in single datasets at the initial stage in analysis we then performed extensive further analyses to explore how these changes related to different data sets and covariates / mechanisms using DIABLO and Spearman's correlations in an effort to assess which covariates / mechanisms influenced which aspects of the microbiota / metabolome.

We have now also added additional results to the manuscript having further isolated the effects of different covariates / mechanisms after adjusting for them, as suggested by the reviewer. We have performed this by adding partial correlation analysis to the manuscript results. In the analysis of correlations to HbA1c we have adjusted for weight and BMI to help isolate weight-independent markers of glycaemic control using partial correlations. Similarly, we have performed partial correlations to BMI adjusting for HbA1c to help reduce the influence of changes to diabetic control after weight loss. We thank the reviewer for highlighting this point and hope that this addition has strengthened the manuscript.

Reviewer: No further comments, I agree with authors comments

a-6) In lines 168-169, Peptostreptococcaceae and Barnesiella are family level. Also, why the associations are reported at the family level, and species level is more appropriate, as this study has done metagenomics and also species at a genus level even have different functions and behavior.

Response:

The analyses were performed at all taxonomic levels including species and genus level. In the manuscript changes predominantly described at the genus level. This is firstly to improve the clarity of the results. If all the species were outlined this would be extremely long. In addition, while metagenomics is purported to allow species-level analysis it is becoming clear from recent studies that this is not the case when using tools such as Kraken, MetaPhlAn and Centrifuge. That is, these tools often do not differentiate closely related species and count them more than once, leading to inflated species-level abundance data (e.g. PMID:32436839). Consequently, we have described the data at genus level, which allowed us to demonstrate differences between our groups of interest.

Regarding the specific points of Peptostreptococcaceae and Barnesiella. Barnesiella is at the genus taxonomic level. Peptostreptococcaceae in this context represented a novel unclassified genus. This has been clarified in the text as Peptostreptococcaceae unclassified.

Reviewer: No further comments, thank you for clarification.

a-7) for pathway enrichment analysis using in house method, is not clear how it works. An option would be to use the coefficients described in a-5 and be used via methods for enrichment analysis. Otherwise, clarification on the in-house method would be sufficient.

Response:

Functional annotation to KEGG pathways was carried out by mapping centroid sequences to the eggNOG-Mapper database. Integration of datasets (e.g. metabolite, microbiota, KEGG etc) was performed using the Data Integration Analysis for Biomarker discovery using Latent cOmponents (DIABLO)¹, implemented through the mixOmics2 package in R.

Analysis of correlations to BMI and HbA1c was performed using Spearman's correlations with multiple testing corrections and as described above, we have now also added partial Spearman's correlations to the analysis to help adjust for covariates.

1. Singh A et al. DIABLO: an integrative approach for identifying key molecular drivers from multi-omics assays. *Bioinformatics*. Sep 1 2019;35(17):3055-3062.
2. Rohart F et al. mixOmics: An R package for 'omics feature selection and multiple data integration. *PLoS Comput Biol*. Nov 2017;13(11):e1005752.

Reviewer: Please add this clarification in the method section or main text.

a-8) Figure 6 is the most important figure I believe. Integration of microbial species and metabolites can be specific to each group and compared in two rows: one for RYGB and one for VSG. Additional analysis would be to include pathways, metabolites, and microbes that are significantly changed post-surgery for each group.

Response:

We thank the reviewer for this suggestion. We have attempted this analysis, however, splitting the data into these groups significantly reduces the n number and hence the power of the tests. We found far fewer correlations: of those seen many were similar, but we were concerned that any correlations not seen were due to a type 2 statistical error rather than any true differences between the subgroups. In addition, as described above we have added two further panels to Figure 6 to better control for covariates / competing mechanisms using partial correlations. While differences between the bariatric surgery procedures have already been explored using DIABLO in Figure 5 and supplemental R4. There is a danger that adding further analyses overcomplicates the analysis, taking away from the key messages.

Reviewer: No further comments. Thanks for the clarification.

a-9) a summary table of significant changes that have been seen before and what is uniquely found in this study will be very beneficial.

Response:

Given the large breath of metabolic, metagenomic, dietary and clinical data this would be challenging to condense into a summary table form.

The overwhelming disruption to the GM and human metabolome caused by bariatric surgery and its functional importance are only just being defined. Few studies exploring host-microbe interactions have occurred in human cohorts, with most studies focussing on either the microbiome or the metabolome. No studies have integrated multi-biofluid analysis to give a comprehensive metabolic insight and importantly none have analysed urine, which gives important insights into gut-host co-metabolism following bariatric surgery. In this study we have integrated extensive multi-biofluid metabolic analyses across multiple biological matrices with shotgun metagenomics (taxonomic and functional profiling) of the GM in both i) T2D versus non-diabetic BMI matched participants at baseline and ii) participants longitudinally profiled pre and post RYGB and vertical sleeve gastrectomy (VSG). This integrative analysis has provided a comprehensive atlas of host-microbe responses to bariatric surgery, glycaemic control and BMI at the systems level and unique insights into the influence of each in humans. Profiling patients with and without T2D has enabled us to dissect the effects of weight-loss over improved glycaemic control. Additionally, we found significant differences between the contrasting bariatric procedures. We identified a number of bacterial taxa and metabolic pathways that are altered following RYGB, but not VSG. These mainly related to increased amino acid metabolism pathways within the GM, with corresponding increases in urinary metabolites associated with microbial protein degradation. Whereas changes in BA metabolism did

not appear to be primarily influenced by altered GM.

As stated by reviewer 3 we hope that: “The findings are of significant interest to the microbiome community as well as those scientific investigators working to identify the underlying pathways that are responsible for the dramatic metabolic improvements following bariatric surgery. This is a large amount of data and will surely generate numerous hypotheses that can potentially be tested in additional clinical and pre-clinical models.”

Reviewer: No further comments,

b) minor:

b-1) In the abstract GM need to be spelled out: “GM-host co-metabolism”.

Response:

This has been amended.

Reviewer: No further comments, thank you.

b-2) In Figure 5, keep the order of subplots in row 1 and row 2 the same for ease of comparison. Also, please make labels (e.g., taxa and KEGG) larger.

Response:

Having row 2 read from right to left allows the main plots (C+F) to be larger and improve readability. However, we would be happy to be guided by the editorial team if they agree that row 2 should read from left to right.

Reviewer: No further comments.

b3) Does the order of phylum to species need to be reversed in Figure 3C.

Response:

The labels represent the nodes in the figure and are correctly ordered.

Reviewer: No further comments.

b4) In figure 4C-D, Y-axis labels need to be revised: $-\log_{10}(q\text{-value}) * (\text{sign of } B)$. sign of B is basically $(B/|B|)$.

Response:

The labels have been revised accordingly.

Reviewer: No further comments.

b5) I would bring a shorter version of Fig 21 and 22 in the main text.

Response:

On the suggestion of the reviewer, we have brought Figure 21 into the main text.

Reviewer: No further comments.

Reviewer #3 (Remarks to the Author):

I thank the authors for the changes that they've made the manuscript. It is much more clear to the

reader. The revised manuscript is much better than the initial submission.

Overall, this is an association study that does not provide much mechanistic evidence – this is a severe limitation of the study, but the study overall has many benefits as described by the reviewers and noted by the authors in the manuscript. I have no further comments.

Reviewer #4 (Remarks to the Author):

The revised version addressed all the issues raised in my previous review. I believe that the paper should be accepted.